# Provably Efficient Reward-Agnostic Navigation with Linear Value Iteration

**Andrea Zanette**
Stanford University
zanette@stanford.edu

**Alessandro Lazaric**
Facebook Artificial Intelligence Research
lazaric@fb.com

**Mykel J. Kochenderfer**
Stanford University
mykel@stanford.edu

**Emma Brunskill**
Stanford University
ebrun@cs.stanford.edu

## Abstract

There has been growing progress on theoretical analyses for provably efficient learning in MDPs with linear function approximation, but much of the existing work has made strong assumptions to enable exploration by conventional exploration frameworks. Typically these assumptions are stronger than what is needed to find good solutions in the batch setting. In this work, we show how under a more standard notion of low inherent Bellman error, typically employed in least-square value iteration-style algorithms, we can provide strong PAC guarantees on learning a near optimal value function provided that the linear space is sufficiently "explorable". We present a computationally tractable algorithm for the reward-free setting and show how it can be used to learn a near optimal policy for any (linear) reward function, which is revealed only once learning has completed. If this reward function is also estimated from the samples gathered during pure exploration, our results also provide same-order PAC guarantees on the performance of the resulting policy for this setting.

## 1 Introduction

Reinforcement learning (RL) aims to solve complex multi-step decision problems with stochastic outcomes framed as a Markov decision process (MDP). RL algorithms often need to explore large state and action spaces where function approximations become necessity. In this work, we focus on exploration with linear predictors for the action value function, which can be quite expressive [Sutton and Barto, 2018].

**Existing guarantees for linear value functions**  Exploration has been widely studied in the tabular setting [Azar et al., 2017, Zanette and Brunskill, 2019, Efroni et al., 2019, Jin et al., 2018, Dann et al., 2019], but obtaining formal guarantees for exploration with function approximation appears to be a challenge even in the linear case. The minimal necessary and sufficient conditions to reliably learn a linear predictor are not fully understood *even with access to a generative model* [Du et al., 2019b]. We know that when the best policy is unique and the predictor is sufficiently accurate it can be identified [Du et al., 2019c, 2020], but in general we are interested in finding only near-optimal policies using potentially misspecified approximators.

To achieve this goal, several ideas from tabular exploration and linear bandits [Lattimore and Szepesvári, 2020] have been combined to obtain provably efficient algorithms in low-rank MDPs [Yang and Wang, 2020, Zanette et al., 2020a, Jin et al., 2020b] and their extension [Wang et al., 2019, 2020b]. We shall identify the core assumption of the above works as *optimistic closure*: all these

settings assume the *Bellman operator maps any value function of the learner to a low-dimensional space $\mathcal{Q}$* that the learner knows. When this property holds, we can add exploration bonuses because *by assumption the Bellman operator maps the agent's optimistically modified value function back to $\mathcal{Q}$*, which the algorithm can represent and use to propagate the optimism and drive the exploration. However, the optimistic closure is put as an assumption to enable *exploration* using traditional methods, but is stronger that what is typically required in the batch setting.

**Towards batch assumptions**   This work is motivated by the desire to have exploration algorithms that we can deploy under more mainstream assumptions, ideally when we can apply well-known *batch* procedures like least square policy iteration (LSPI) [Lagoudakis and Parr, 2003], and least square value iteration (LSVI) [Munos, 2005].

LSPI has convergence guarantees when the action value function of *all* policies can be approximated with a linear architecture [Lazaric et al., 2012], i.e., $Q^\pi$ is linear for all $\pi$; in this setting, Lattimore and Szepesvari [2020] recently use a design-of-experiments procedure from the bandit literature to obtain a provably efficient algorithm for finding a near optimal policy, but they need access to a generative model. LSVI, another popular batch algorithm, requires low inherent Bellman error [Munos and Szepesvári, 2008, Chen and Jiang, 2019]. In this setting, Zanette et al. [2020b] present a near-optimal (with respect to noise and misspecification) regret-minimizing algorithm that operates online, but a computationally tractable implementation is not known. It is worth noting that both settings are more general than linear MDPs [Zanette et al. 2020b].

A separate line of research is investigating settings with low Bellman rank [Jiang et al. 2017] which was found to be a suitable measure of the learnability of many complex reinforcement learning problems. The notion of Bellman rank extends well beyond the linear setting.

The lack of computational tractability in the setting of Zanette et al. [2020b] and in the setting with low Bellman rank [Jiang et al. 2017] and of a proper online algorithm in [Lattimore and Szepesvari, 2020] highlight the hardness of these very general settings which do not posit additional assumptions on the linear value function class $\mathcal{Q}$ beyond what is required in the batch setting.

**Reward-free exploration**   We tackle the problem of designing an exploration algorithm using batch assumptions by adopting a pure exploration perspective: our algorithm can return a near optimal policy for any linear reward function that is revealed after an initial learning phase. It is therefore a probably approximately correct (PAC) algorithm. Reward-free exploration has been investigated in the tabular setting with an end-to-end algorithm [Jin et al., 2020a]. Hazan et al. [2018] design an algorithm for a more general setting through oracles that also recovers guarantees in the tabular domains. Others [Du et al., 2019a, Misra et al., 2020] also adopt the pure exploration perspective assuming a small but unobservable state space. More recently, reward free exploration has gained attention in the tabular setting Kaufmann et al. [2020], Tarbouriech et al., Ménard et al. [2020] as well as the context of function approximation Wainwright [2019], Agarwal et al. [2020].

**Contribution**   This works makes two contributions. It presents a statistically and computationally efficient online PAC algorithm to learn a near-optimal policy 1) for the setting with low inherent Bellman error [Munos and Szepesvári, 2008] and 2) for reward-free exploration in the same setting.

From a technical standpoint, 1) implies we cannot use traditional exploration methodologies and 2) implies we cannot learn the full dynamics, which would require estimating all state-action-state transition models. Both goals are accomplished by driving exploration by approximating G-optimal experimental design [Lattimore and Szepesvári, 2020] in online reinforcement learning through randomization. Our algorithm returns a dataset of well chosen state-action-transition triplets, such that invoking the LSVI algorithm on that dataset (with a chosen reward function) returns a near optimal policy on the MDP with that reward function.

## 2   Preliminaries and Intuition

We consider an undiscounted $H$-horizon MDP [Puterman, 1994] $M = (\mathcal{S}, \mathcal{A}, p, r, H)$ defined by a possibly infinite state space $\mathcal{S}$ and action space $\mathcal{A}$. For every $t \in [H] = \{1, \ldots, H\}$ and state-action pair $(s, a)$, we have a reward function $r_t(s, a)$ and a transition kernel $p_t(\cdot \mid s, a)$ over the next state. A policy $\pi$ maps a $(s, a, t)$ triplet to an action and defines a reward-dependent action value function

$Q_t^\pi(s,a) = r_t(s,a) + \mathbb{E}\left[\sum_{l=t+1}^H r_l(s_l, \pi_l(s_l)) \mid s, a\right]$ and a value function $V_t^\pi(s) = Q_t^\pi(s, \pi_t(s))$. For a given reward function there exists an optimal policy $\pi^\star$ whose value and action-value functions on that reward function are defined as $V_t^\star(s) = \sup_\pi V_t^\pi(s)$ and $Q_t^\star(s,a) = \sup_\pi Q_t^\pi(s,a)$. We indicate with $\rho$ the starting distribution. The Bellman operator $\mathcal{T}_t$ applied to the action value function $Q_{t+1}$ is defined as $\mathcal{T}_t(Q_{t+1})(s,a) = r_t(s,a) + \mathbb{E}_{s'\sim p_t(s,a)} \max_{a'} Q_{t+1}(s',a')$. For a symmetric positive definite matrix $\Sigma$ and a vector $x$ we define $\|x\|_{\Sigma^{-1}} = \sqrt{x^\top \Sigma^{-1} x}$. The $O(\cdot)$ notation hides constant values and the $\widetilde{O}(\cdot)$ notation hides constants and $\ln(dH\frac{1}{\epsilon}\frac{1}{\delta})$, where $d$ is the feature dimensionality described next.

**Linear Approximators** For the rest of the paper we restrict our attention to linear functional spaces for the action value function, i.e., where $Q_t(s,a) \approx \phi_t(s,a)^\top \theta$ for a known feature extractor $\phi_t(s,a)$ and a parameter $\theta$ in a certain set $\mathcal{B}_t$, which we assume to be the Euclidean ball with unit radius $\mathcal{B}_t = \{\theta \in \mathbb{R}^{d_t} \mid \|\theta\|_2 \leq 1\}$. This defines the value functional spaces as

$$\mathcal{Q}_t \stackrel{def}{=} \{Q_t \mid Q_t(s,a) = \phi_t(s,a)^\top\theta, \ \theta \in \mathcal{B}_t\}, \quad \mathcal{V}_t \stackrel{def}{=} \{V_t \mid V_t(s) = \max_a \phi_t(s,a)^\top\theta, \ \theta \in \mathcal{B}_t\}.$$

**Inherent Bellman error** The inherent Bellman error condition is typically employed in the analysis of LSVI [Munos and Szepesvári, 2008, Chen and Jiang, 2019]. It measures the closure of the prescribed functional space $\mathcal{Q}$ with respect to the Bellman operator $\mathcal{T}$, i.e, the distance of $\mathcal{T}Q$ from $\mathcal{Q}$ *provided that* $Q \in \mathcal{Q}$. In other words, low inherent Bellman error ensures that if we start with an action value function in $\mathcal{Q}$ then we approximately remain in the space after performance of the Bellman update. For finite horizon MDP we can define the inherent Bellman error as:

$$\max_{Q_{t+1}\in\mathcal{Q}_{t+1}} \min_{Q_t\in\mathcal{Q}_t} \max_{(s,a)} |[Q_t - \mathcal{T}_t(Q_{t+1})](s,a)|. \tag{1}$$

When linear function approximations are used and the inherent Bellman error is zero, we are in a setting of low Bellman rank [Jiang et al., 2017], where the Bellman rank is the feature dimensionality. This condition is more general than the low rank MDP setting or optimistic closure [Yang and Wang, 2020, Jin et al., 2020b, Zanette et al., 2020a, Wang et al., 2019]; for a discussion of this see [Zanette et al., 2020b].

**Model-free reward-free learning** In the absence of reward signal, how should $\mathcal{Q}_t$ look like? Define the reward-free Bellman operator $\mathcal{T}_t^P(Q_{t+1})(s,a) = \mathbb{E}_{s'\sim p_t(s,a)} \max_{a'} Q_{t+1}(s',a')$. It is essentially equivalent to measure the Bellman error either on the full Bellman operator $\mathcal{T}_t$ or directly on the dynamics $\mathcal{T}_t^P$ when the reward function is linear (see proposition 2 of Zanette et al. [2020b]). We therefore define the inherent Bellman error directly in the transition operator $\mathcal{T}^P$:

**Definition 1** (Inherent Bellman Error).

$$\mathcal{I}(\mathcal{Q}_t, \mathcal{Q}_t) \stackrel{def}{=} \max_{Q_{t+1}\in\mathcal{Q}_{t+1}} \min_{Q_t\in\mathcal{Q}_t} \max_{(s,a)} |Q_t - \mathcal{T}_t^P(Q_{t+1})](s,a)|. \tag{2}$$

**Approximating G-optimal design** G-optimal design is a procedure [Kiefer and Wolfowitz, 1960] that identifies an appropriate sequence of features $\phi_1, \dots \phi_n$ to probe to form the design matrix $\Sigma = \sum_{i=1}^n \phi_i \phi_i^\top$ in order to uniformly reduce the maximum "uncertainty" over all the features as measured by $\max_\phi \|\phi\|_{\Sigma^{-1}}$, see appendix C. This principle has recently been applied to RL with a generative model [Lattimore and Szepesvari, 2020] to find a near optimal policy.

However, the basic idea has the following drawbacks in RL: 1) it requires access to a generative model; 2) it is prohibitively expensive as it needs to examine all the features across the full state-action space before identifying what features to probe. This work addresses these 2 drawbacks in reinforcement learning by doing two successive approximations to G-optimal design. The first approximation would be compute and follow the policy $\pi$ (different in every rollout) that leads to an expected feature $\overline{\phi}_\pi$ in the most uncertain direction[1] (i.e., the direction where we have the least amount of data). This solves problem 1 and 3 above, but unfortunately it turns out that computing such $\pi$ is computationally infeasible. Thus we relax this program by finding a policy that in most of the episodes makes at least some progress in the most uncertain direction, thereby addressing point 2 above. This is achieved through randomization; the connection is briefly outlined in section 5.5.

# 3 Algorithm

Moving from the high-level intuition to the actual algorithm requires some justification, which is left to section 5. Here instead we give few remarks about algorithm 1: first, the algorithm proceeds in phases $p = 1, 2, \ldots$ and in each phase it focuses on learning the corresponding timestep (e.g., in phase 2 it learns the dynamics at timestep 2). Proceeding forward in time is important because *to explore at timestep $p$ the algorithm needs to know how to navigate through prior timesteps*. Second, we found that random sampling a reward signal in the exploratory timestep from the inverse covariance matrix $\xi_p \sim \mathcal{N}(0, \sigma \Sigma_{pk}^{-1})$ is an elegant and effective way to *approximate design of experiment* (see section 5.5), although this is not the only possible choice. Variations of this basic protocol are broadly known in the literature as Thompson sampling [Osband et al., 2016a, Agrawal and Jia, 2017, Russo, 2019, Gopalan and Mannor, 2015, Ouyang et al., 2017] and from an algorithmic standpoint our procedure could be interpreted as a modification of the popular RLSVI algorithm [Osband et al., 2016b] to tackle the reward-free exploration problem.

---

**Algorithm 1** *Forward Reward Agnostic Navigation with Confidence by Injecting Stochasticity* (FRANCIS)

---

1: **Inputs**: failure probability $\delta \in [0, 1]$, target precision $\epsilon > 0$, feature map $\phi$
2: Initialize $\Sigma_{t1} = \lambda I, \widehat{\theta}_t = 0, \forall t \in [H], \mathcal{D} = \emptyset$; set $c_e, c_\sigma, c_\alpha \in \mathbb{R}$ (see appendix), $\lambda = 1$
3: **for** phase $p = 1, 2, \ldots, H$ **do**
4:     $k = 1$, set $\sigma = \sigma_{start} \overset{def}{=} c_\sigma / (d_p \ln(\frac{d_p}{\delta \epsilon}))$
5:     **while** $\sigma < c_\alpha H^2 (d_p + d_{p+1}) \ln(\frac{d_p}{\epsilon \delta})$ **do**
6:         **for** $i = 1, 2, \ldots, c_e \frac{d_p^2 \sigma}{\epsilon^2}$ **do**
7:             $k = k + 1$, receive starting state $s_1 \sim \rho$
8:             $\xi_p \sim \mathcal{N}(0, \sigma \Sigma_{pk}^{-1})$; $\quad \mathrm{R}_p(s, a) \overset{def}{=} \phi_p(s, a)^\top \xi_p$
9:             $\pi \longleftarrow \mathrm{LSVI}(p, \mathrm{R}_p, \mathcal{D})$
10:            Run $\pi$; $\mathcal{D} \leftarrow \mathcal{D} \cup (s_{pk}, a_{pk}, s_{p+1,k}^+)$;
11:            $\phi_{pk} \overset{def}{=} \phi_p(s_{pk}, a_{pk})$; $\Sigma_{p,k+1} \leftarrow \Sigma_{pk} + \phi_{pk}\phi_{pk}^\top$
12:        **end for**
13:        $\sigma \longleftarrow 2\sigma$
14:    **end while**
15: **end for**
16: **return** $\mathcal{D}$

---

The algorithm returns a dataset $\mathcal{D}$ of well chosen state-action-transitions approximating a G-optimal design in the online setting; the dataset can be augmented with the chosen reward function and used in LSVI (detailed in appendix B) to find a near-optimal policy on the MDP with that reward function. The call $\mathrm{LSVI}(p, \mathrm{R}_p, \mathcal{D})$ invokes the LSVI algorithm on a $p$ horizon MDP on the batch data $\mathcal{D}$ with reward function $\mathrm{R}_p$ at timestep $p$.

# 4 Main Result

Before presenting the main result is useful to define the average feature $\overline{\phi}_{\pi,t} = \mathbb{E}_{x_t \sim \pi} \phi_t(x_t, \pi_t(x_t))$ encountered at timestep $t$ upon following a certain policy $\pi$. In addition, we need a way to measure how "explorable" the space is, i.e., how easy it is to collect information in a given direction of the feature space using an appropriate policy. The explorability coefficient $\nu$ measures how much we can align the expected feature $\overline{\phi}_{\pi,t}$ with the most challenging direction $\theta$ to explore even if we use the *best* policy $\pi$ for the task (i.e., the policy that maximizes this alignment). It measures how difficult it is to explore the most challenging direction, even if we use the best (and usually unknown) policy to do so. This is similar to a diameter condition in the work of Jaksch et al. [2010] in the features space, but different from ergodicity, which ensures that sufficient information can be collected by *any* policy. It is similar to the reachability parameter of Du et al. [2019a] and Misra et al. [2020], but our condition concerns the features rather than the state space and is unavoidable in certain settings (see discussion after the main theorem).

**Definition 2** (Explorability). $\nu_t \overset{def}{=} \min_{\|\theta\|_2=1} \max_\pi |\overline{\phi}_{\pi,t}^\top \theta|; \qquad \nu_{min} = \min_{t \in [H]} \nu_t$.

**Theorem 4.1.** *Assume $\|\phi_t(s, a)\|_2 \leq 1$ and set $\epsilon$ to satisfy $\epsilon \geq \widetilde{O}(d_t H \mathcal{I}(\mathcal{Q}_t, \mathcal{Q}_{t+1}))$ and $\epsilon \leq \widetilde{O}(\nu_{min}/\sqrt{d_t})$ for all $t \in [H]$. FRANCIS terminates after $\widetilde{O}\left(H^2 \sum_{t=1}^H \frac{d_t^2(d_t + d_{t+1})}{\epsilon^2}\right)$ episodes.*

*Fix a reward function $r_t(\cdot, \cdot)$ such that each state-action-successor state $(s_{tk}, a_{tk}, s_{t+1,k}^+)$ triplet in $\mathcal{D}$ (where $t \in [H]$ and $k$ is the episode index in phase $t$) is augmented with a reward $r_{tk} = r_t(s_{tk}, a_{tk})$.*

| | Online? | Reward-agnostic? | Need optimistic closure? | # episodes | # computations |
|---|---|---|---|---|---|
| This work | Yes | Yes | No | $\frac{d^3 H^5}{\epsilon^2}$ | poly$(d, H, 1/\epsilon^2)$ |
| G-optimal design + LSVI | No | Yes | No | $\frac{d^2 H^5}{\epsilon^2}$ | $\Omega(SA)$ |
| [Zanette et al., 2020b] | Yes | No | No | $\frac{d^2 H^4}{\epsilon^2}$ | exponential |
| [Jin et al., 2020b] | Yes | No | Yes | $\frac{d^3 H^4}{\epsilon^2}$ | poly$(d, H, 1/\epsilon^2)$ |
| [Jiang et al., 2017] | Yes | No | No | $\frac{d^2 H^5}{\epsilon^2}|\mathcal{A}|$ | intractable |
| [Jin et al., 2020a] | Yes | Yes | (tabular) | $\frac{H^5 S^2 A}{\epsilon^2}$ | poly$(S, A, H, 1/\epsilon^2)$ |
| [Wang et al., 2020a] | Yes | Yes | Yes | $\frac{d^3 H^6}{\epsilon^2}$ | poly$(S, A, H, 1/\epsilon^2)$ |

Table 1: We consider the number of episodes to learn an $\epsilon$-optimal policy. We assume $r \in [0, 1]$ and $Q^\pi \in [0, H]$, and rescale the results to hold in this setting. We neglect misspecification for all works. The column "optimistic closure" refers to the assumption that the Bellman operator projects *any* value function into a prescribed space (notably, low-rank MDPs of [Jin et al., 2020b]). For our work we assume $\epsilon = \Omega(\nu_{min}/\sqrt{d})$. We recall that if an algorithm has regret $A\sqrt{K}$, with $K$ the number of episodes then we can extract a PAC algorithm to return an $\epsilon$-optimal policy in $\frac{A^2}{\epsilon^2}$ episodes. We evaluate [Jiang et al., 2017] in our setting where the Bellman rank is $d$ (the result has an explicit dependence on the number of actions, though this could be improved in the linear setting). $G$-optimal design is from the paper [Lattimore and Szepesvari, 2020] which operates in infinite-horizon and assuming linearity of $Q^\pi$ for all $\pi$, so the same idea of G-optimal design was applied to our setting to derive the result and we report the number of required samples (as opposed to the number of episodes), see appendix C. For [Jin et al., 2020a] we ignore the $\frac{H^7 S^4 A}{\epsilon}$ lower order term

*If the reward function $r_t(\cdot, \cdot)$ satisfies for some parameters $\theta_1^r \in \mathbb{R}^{d_1}, \dots, \theta_H^r \in \mathbb{R}^{d_H}$*

$$\forall (s, a, t) \quad \|\theta_t^r\|_2 \leq \frac{1}{H}, \quad r_t(s, a) = \phi_t(s, a)^\top \theta_t^r$$

*then with probability at least $1 - \delta$ the policy $\pi$ returned by* LSVI *using the augmented dataset $\mathcal{D}$ satisfies (on the MDP with $r_t(\cdot, \cdot)$ as reward function)*

$$\mathbb{E}_{x_1 \sim \rho}(V_1^\star - V_1^\pi)(x_1) \leq \epsilon. \tag{3}$$

The full statement is reported in appendix appendix D.6. The reward function $r_t(\cdot, \cdot)$ could even be adversarially chosen after the algorithm has terminated. If the reward function is estimated from data then the theorem immediately gives same-order guarantees as a corollary. The dynamics error $O(d_t H \mathcal{I}(\mathcal{Q}_t, \mathcal{Q}_{t+1}))$ is contained in $\epsilon$.

The setting allows us to model MDPs where where $r_t \in [0, \frac{1}{H}]$ and $V_t^\star \in [0, 1]$. When applied to MDPs with rewards in $[0, 1]$ (and value functions in $[0, H]$), the input and output should be rescaled and the number of episodes to $\epsilon$ accuracy should be multiplied by $H^2$.

The significance of the result lies in the fact that this is the first statistically and computationally[2] efficient PAC algorithm for the setting of low inherent Bellman error; this is special case of the setting with low Bellman rank (the Bellman rank being the dimensionality of the features). In addition, this work provides one of the first end-to-end algorithms for provably efficient reward-free exploration with linear function approximation.

In table 1 we describe our relation with few relevant papers in the field. The purpose of the comparison is not to list the pros and cons of each work with respect to one another, as these works all operate under different assumptions, but rather to highlight what is achievable in different settings.

**Is small Bellman error needed?** As of writing, the minimal conditions that enable provably efficient learning with function approximation are still unknown [Du et al., 2019b]. In this work we focus on small Bellman error which is a condition typically used for *batch* analysis of LSVI [Munos, 2005, Munos and Szepesvári, 2008, Chen and Jiang, 2019]. What is really needed for the functioning of FRANCIS is that vanilla LSVI outputs a good solution in the limit of infinite data on different (linear) reward functions: as long as LSVI can return a near-optimal policy for the given reward function given enough data, FRANCIS can proceed with the exploration. This requirement

is really minimal, because even if the best dataset $\mathcal{D}$ is collected through G-optimal design on a generative model (instead of using FRANCIS), LSVI must anyway be able to output a good policy on the prescribed reward function.

**Is explorability needed?** Theorem 4.1 requires $\epsilon \leq \widetilde{O}(\nu_{min}/\sqrt{d_t})$. Unfortunately, a dependence on $\nu_{min}$ turns out to be unavoidable in the *more general* setting we consider in the appendix; we discuss this in more detail in appendix E, but here we give some intuition regarding the explorability requirement.

FRANCIS can operate under two separate set of assumptions, which we call *implicit* and *explicit* regularity, see definition 6 (*Reward Classes*) in appendix and the main result in theorem 1.

Under *implicit* regularity we do not put assumptions on the norm of reward parameter $\|\theta^r\|_2$, but only a bound on the expected value of the rewards under any policy: $|\mathbb{E}_{x_t \sim \pi} r_t(x_t, \pi_t(x_t))| \leq \frac{1}{H}$. This representation allows us to represent *very high rewards* ($\gg 1$) *in hard-to-reach states*. It basically controls how big the value function can get. This setting is more challenging for an agent to explore *even in the tabular setting* and *even in the case of a single reward function*. If a state is hard to reach, the reward there can be very high, and a policy that tries to go there *can still have high value*. Under this implicit regularity assumption, the explorability parameter would show up for tabular algorithms as well (as minimum visit probability to any state under an appropriate policy).

By contrast, under *explicit* regularity (which concerns the result reported in theorem 4.1) we do make the classical assumption that bounds the parameter norm $\|\theta^r\|_2 \leq 1/H$. In this case, the lower bound no longer applies, but the proposed algorithm still requires good "explorability" to proceed. Removing this assumption is left as future work.

# 5 Technical Analysis

For the proof sketch we neglect misspecification, i.e., $\mathcal{I}(\mathcal{Q}_t, \mathcal{Q}_{t+1}) = 0$. We say that a statement holds with very high probability if the probability that it does not hold is $\ll \delta$.

## 5.1 Analysis of LSVI, uncertainty and inductive hypothesis

FRANCIS repeatedly calls LSVI on different randomized linearly-parameterized reward functions $R_p$ and so we need to understand how the signal propagates. Let us begin by defining an uncertainty function in episode $i$ of phase $p$ using the covariance matrix $\Sigma_{pi} = \sum_{j=1}^{i-1} \phi_{pj} \phi_{pj}^\top + I$ on the observed features $\phi_{pj} = \phi_p(s_{pj}, a_{pj})$ at episode $j$ of phase $p$:

**Definition 3** (Max Uncertainty). $\mathcal{U}_{pi}^\star(\sigma) \overset{def}{=} \max_{\pi, \|\theta^\mathcal{U}\|_{\Sigma_{pi}} \leq \sqrt{\sigma}} \overline{\phi}_{\pi,p}^\top \theta^\mathcal{U} \overset{def}{=} \max_\pi \sqrt{\sigma} \|\overline{\phi}_{\pi,p}\|_{\Sigma_{pi}^{-1}}$.

Let $\Sigma_t$ denote the covariance matrix in timestep $t$ once learning in that phase has completed, and likewise denote with $\mathcal{U}_t^\star(\sigma)$ the final value of the program of definition 3 once learning in phase $t$ has completed (so using $\Sigma_t$ in the definition); let $\sqrt{\alpha_t} = \widetilde{O}(\sqrt{d_t + d_{t+1}})$ and $R_p(s,a) = \phi_p(s,a)^\top \xi_p$.

**Lemma 1** (see appendix B.4). *Assume $\|\xi_p\|_2 \leq 1$ and $\lambda_{min}(\Sigma_t) = \Omega(H^2 \alpha_t)$ for all $t \in [p-1]$. Then with very high probability LSVI$(p, R_p, \mathcal{D})$ computes a value function $\widehat{V}$ and a policy $\pi$ s.t.*

$$|\mathbb{E}_{x_1 \sim \rho} \widehat{V}_1(x_1) - \overline{\phi}_{\pi,p}^\top \xi_p| \leq \sum_{t=1}^{p-1} \left[ \sqrt{\alpha_t} \|\overline{\phi}_{\pi,t}\|_{\Sigma_t^{-1}} \right] = \sum_{t=1}^{p-1} \mathcal{U}_t^\star(\alpha_t) = \text{Least-Square Error}.$$

The least-square error in the above display can be interpreted as a planning error to propagate the signal $\xi_p$; it also appears when LSVI uses the batch dataset $\mathcal{D}$ to find the optimal policy on a given reward function after FRANCIS has terminated, and it is the quantity we target to reduce. Since $\alpha_t$ is constant, we need to shrink $\|\overline{\phi}_{\pi,p}\|_{\Sigma_p^{-1}}$ over any choice of $\pi$ as much as possible by obtaining an appropriate[3] feature matrix $\Sigma_t$.

A final error across all timesteps of order $\epsilon$ can be achieved when the algorithm adds at most $\epsilon/H$ error at every timestep. Towards this, we define an inductive hypothsis that the algorithm has been successful up to the beginning of phase $p$ in reducing the uncertainty encoded in $\mathcal{U}_t^\star$:

**Inductive Hypothesis 1.** *At the start of phase $p$ we have $\sum_{t=1}^{p-1} \mathcal{U}_t^\star(\alpha_t) \leq \frac{p-1}{H}\epsilon$.*

The inductive hypothesis *critically ensures* that the reward signal $\xi$ can be accurately propagated backward by LSVI, enabling navigation capabilities of FRANCIS to regions of uncertainty in phase $p$ (this justifies the phased design of FRANCIS).

### 5.2 Overestimating the maximum uncertainty through randomization

Assuming the inductive hypothesis, we want to show how to reduce the uncertainty in timestep $p$. Similar to how optimistic algorithms overestimate the optimal value function, here $\mathbb{E}_{x_1 \sim \rho} \widehat{V}_1(x_t) \approx \overline{\phi}_{\pi,p}^\top \xi_p$ should overestimate the current uncertainty in episode $i$ of phase $p$ encoded in $\mathcal{U}_{pi}^\star(\alpha_p)$. This is achieved by introducing a randomized reward signal $\xi_{pi} \sim \mathcal{N}(0, \sigma\Sigma_{pi}^{-1})$ at timestep $p$.

**Lemma 2** (Uncertainty Overestimation, appendix D.2). *If $\xi_p \sim \mathcal{N}(0, \sigma\Sigma_{pi}^{-1})$, $\mathcal{U}_{pi}^\star(\sigma) = \Omega(\epsilon)$, $\|\xi_p\|_2 \leq 1$ and the inductive hypothesis holds then LSVI returns with some constant probability $q \in \mathbb{R}$ a policy $\pi$ such that $\overline{\phi}_{\pi,p}^\top \xi_{pi} \geq \mathcal{U}_{pi}^\star(\sigma)$.*

The proof of the above lemma uses lemma 1. The condition $\mathcal{U}_{pi}^\star(\sigma) = \Omega(\epsilon)$ is needed: if the signal $\xi_{pi}$ or uncertainty $\mathcal{U}_{pi}^\star(\sigma)$ are too small relative to $\epsilon$ then the least-square error of order $\epsilon$ that occurs in LSVI is too large relative to the signal $\xi_{pi}$, and the signal cannot be propagated backwardly.

The lemma suggests we set $\sigma = \alpha_t$ to ensure $\overline{\phi}_{\pi,p}^\top \xi_{pi} \geq \mathcal{U}_{pi}^\star(\alpha_t)$ with fixed probability $q \in \mathbb{R}$. Unfortunately this choice would generate a very large $\|\xi_{pi}\|_2$ which violates the condition $\|\xi_{pi}\|_2 \leq 1$. In particular, the condition $\|\xi_{pi}\|_2 \leq 1$ determines how big $\sigma$ can be.

**Lemma 3** (see appendix D.1). *If $\sigma = \widetilde{O}(\lambda_{min}(\Sigma_{pi})/d_p)$ and $\xi_{pi} \sim \mathcal{N}(0, \sigma\Sigma_{pi}^{-1})$ then $\|\xi_{pi}\|_2 \leq 1$ with very high probability.*

Since initially $\Sigma_{p1} = I$, the above lemma determines the initial value $\sigma \approx 1/d_p \ll \alpha_p$. This implies FRANCIS won't be able to overestimate the uncertainty $\mathcal{U}_{pi}^\star(\alpha_t)$ initially.

The solution is to have the algorithm proceed in epochs. At the end of every epoch FRANCIS ensures $\mathcal{U}_{pi}^\star(\sigma) \leq \epsilon$, and that $\lambda_{min}(\Sigma_{pi})$ is large enough that $\sigma$ can be doubled at the beginning of the next epoch.

### 5.3 Learning an Epoch

Using lemma 2 we can analyze what happens within an epoch when $\sigma$ is fixed (assuming $\sigma$ is appropriately chosen to ensure $\|\xi_p\|_2 \leq 1$ with very high probability). We first consider the average uncertainty as a measure of progress and derive the bound below by neglecting the small error from encountering the feature $\phi_{pi}$ (step $(a)$ below) instead of the expected feature $\overline{\phi}_{\pi_i,p}$ (identified by the policy $\pi_i$ played by FRANCIS in episode $i$), by using a high probability bound $\|\xi_{pi}\|_{\Sigma_{pi}} \lesssim \sqrt{d_p\sigma}$ and by using the elliptic potential lemma in Abbasi-Yadkori et al. [2011] for the last step.

$$\frac{1}{k}\sum_{i=1}^{k} \mathcal{U}_{pi}^\star(\sigma) \overset{\text{lemma 2}}{\leq} \frac{1}{k}\sum_{i=1}^{k} \overline{\phi}_{\pi_i,p}^\top \xi_{pi} \overset{(a)}{\approx} \frac{1}{k}\sum_{i=1}^{k} \phi_{pi}^\top \xi_{pi} \overset{\substack{\text{Cauchy}\\\text{Schwartz}}}{\leq} \frac{1}{k}\sum_{i=1}^{k} \|\phi_{pi}\|_{\Sigma_{pi}^{-1}} \overbrace{\|\xi_{pi}\|_{\Sigma_{pi}}}^{\lesssim\sqrt{d_p\sigma}} \tag{4}$$

$$\overset{\substack{\text{Cauchy}\\\text{Schwartz}}}{\leq} \sqrt{\frac{d_p\sigma}{k}}\sqrt{\sum_{i=1}^{k} \|\phi_{pi}\|_{\Sigma_{pi}^{-1}}^2} \overset{\substack{\text{Elliptic}\\\text{Pot. Lemma}}}{\leq} d_p\sqrt{\frac{\sigma}{k}}. \tag{5}$$

The inequality $\overline{\phi}_{\pi_i,p}^\top \xi_{pi} \geq \mathcal{U}_{pi}^\star(\sigma)$ in the first step only holds for some of the episodes (since lemma 2 ensures the inequality with probability $q \in \mathbb{R}$), but this only affects the bound up to a constant with high probability. Since the uncertainty is monotonically decreasing, the last term $\mathcal{U}_{pk}^\star(\sigma)$ must be

smaller than the average (the lhs of the above display), and we can conclude $\mathcal{U}^\star_{pk}(\sigma) \leq d_p\sqrt{\sigma/k}$. Asking for the rhs to be $\leq \epsilon$ suggests we need $\approx d_p^2\sigma/\epsilon^2$ episodes. In essence, we have just proved the following:

**Lemma 4** (Number of trajectories to learn an epoch, see appendix D.3). *In a given epoch* FRANCIS *ensures* $\mathcal{U}^\star_{pk}(\sigma) \leq \epsilon$ *with high probability using* $\widetilde{O}(d_p^2\sigma/\epsilon^2)$ *trajectories.*

At the end of an epoch FRANCIS ensures $\mathcal{U}^\star_{pk}(\sigma) \leq \epsilon$, but we really need $\mathcal{U}^\star_{pk}(\alpha_p) \leq \epsilon$ to hold.

### 5.4 Learning a Phase

We need to use the explorability condition to allow FRANCIS to proceed to the next epoch:

**Lemma 5** (see appendix D.4). *Let $\underline{k}$ and $\overline{k}$ be the starting and ending episodes in an epoch. If* $\epsilon = \widetilde{O}(\nu_{min}/\sqrt{d_p})$ *and* $\mathcal{U}^\star_{p\overline{k}}(\sigma) = \widetilde{O}(\epsilon)$ *then* $\lambda_{min}(\Sigma_{p\overline{k}}) \geq 2\lambda_{min}(\Sigma_{p\underline{k}})$.

Since the minimum eigenvalue for the covariance matrix has doubled, we can double $\sigma$ (i.e., inject a stronger signal) and still satisfy lemma 3; at this point FRANCIS enters into a new epoch. At the beginning of every epoch we double $\sigma$, and this is repeated until $\sigma$ reaches the final value $\sigma \approx H^2\alpha_p$. There are therefore only logarithmically many epochs (in the input parameters).

**Lemma 6** (FRANCIS meets target accuracy at the end of a phase, see appendix D.4). *When* FRANCIS *reaches the end of the last epoch in phase $p$ it holds that $\sigma \approx H^2\alpha_p$ and $\epsilon \geq \mathcal{U}^\star_p(\sigma) = H\mathcal{U}^\star_p(\alpha_p)$. This implies $\mathcal{U}^\star_p(\alpha_p) \leq \epsilon/H$, as desired. Furthermore, this is achieved in $\widetilde{O}(d_p^2H^2\alpha_p/\epsilon)$ episodes.*

Since $\mathcal{U}^\star_p(\alpha_p) \leq \epsilon/H$ the inductive step is now proved; summing the number of trajectories over all the phases gives the final bound in theorem 4.1. At this point, an $\epsilon$-optimal policy can be extracted by LSVI on the returned dataset $\mathcal{D}$ for any prescribed linear reward function.

### 5.5 Connection with G-optimal design

We briefly highlight the connection with G-optimal design. G-optimal design would choose a design matrix $\Sigma$ such that $\|\overline{\phi}_{\pi,p}\|_{\Sigma^{-1}}$ is as small as possible for all possible $\pi$. Since we cannot choose the features in the online setting, a first relaxation is to instead compute (and run) the policy $\pi$ that maximizes the program $\mathcal{U}^\star_{pi}(\sigma)$ in every episode $i$. Intuitively, as the area of maximum uncertainty is reached, information is acquired there and the uncertainty is progressively reduced, even though this might be not the most efficient way to proceed from an information-theoretic standpoint. Such procedure would operate in an online fashion, but unfortunately it requires an intractable optimization in policy space. Nonetheless this is the first relaxation to G-optimal design. To obtain the second relaxation, it is useful to consider the alternative definition $\mathcal{U}^\star_{pi}(\sigma) = \max_{\pi,\|\theta^{\mathcal{U}}\|_{\Sigma_{pi}} \leq \sqrt{\sigma}} \overline{\phi}^\top_{\pi,p}\theta^{\mathcal{U}}$. If we relax the constraint $\|\theta^{\mathcal{U}}\|_{\Sigma_{pi}} \leq \sqrt{\sigma}$ to obtain $\|\theta^{\mathcal{U}}\|_{\Sigma_{pi}} \lesssim \sqrt{d_p\sigma}$ then the feasible space is large enough that random sampling from the feasible set (and computing the maximizing policy by using LSVI) achieves the goal of overestimating the maximum of the unrelaxed program; in particular, sampling $\xi_{pi} \sim \mathcal{N}(0, \sigma\Sigma_{pi}^{-1})$ satisfies the relaxed constraints with high probability and is roughly uniformly distributed in the constraint set.

## 6 Discussion

This works makes progress in relaxing the optimistic closure assumptions on the function class for exploration through a statistically and computationally efficient PAC algorithm. From an algorithmic standpoint, our algorithm is inspired by [Osband et al., 2016b], but from an analytical standpoint, it is justified by a design-of-experiments approach [Lattimore and Szepesvari, 2020]. Remarkably, our approximations to make G-experimental design implementable *online* and with polynomial *computational complexity* only add a $d$ factor compared to G-optimal design. The proof technique is new to our knowledge both in principles and in execution, and can be appreciated in the appendix. We hope that the basic principle is general enough to serve as a foundation to develop new algorithms with even more general function approximators. The contribution to reward-free exploration [Jin et al., 2020a] to linear value functions is also a contribution to the field.

# 7 Broader Impact

This work is of theoretical nature and aims at improving our core understanding of reinforcement learning; no immediate societal consequences are anticipated as a result of this study.

## Acknowledgment

Funding in direct support of this work: Total Innovation Program Fellowship, ONR YIP and NSF career. The authors are grateful to the reviewers for their useful comments, in particular about the explorability requirement.

## Footnotes

[1]This is an approximation to $G$-optimal design, because $\pi$ here is the policy that leads to the most uncertain direction $\overline{\phi}_\pi$ rather than to the direction that reduces the uncertainty the most.

[2]FRANCIS requires only polynomial calls to LSVI and samples from a multivariate normal, see appendix D.7.

[3]G-optimal design does this optimally, but requires choosing the features, which is only possible if one has access to a generative model or in a bandit problem.

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
