[Supplementary Material]

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

}_p^{\star}(\sigma) = H\mathcal{U}_p^{\star}(\alpha_p)$. This implies $\mathcal{U}_p^{\star}(\alpha_p) \leq \epsilon/H$, as desired. Furthermore, this is achieved in $\widetilde{O}(d_p^2 H^2\alpha_p/\epsilon)$ episodes.*

Since $\mathcal{U}_p^{\star}(\alpha_p) \leq \epsilon/H$ the inductive step is now proved; summing the number of trajectories over all the phases gives the final bound in theorem 4.1. At this point, an $\epsilon$-optimal policy can be extracted by LSVI on the returned dataset $\mathcal{D}$ for any prescribed linear reward function.

### 5.5 Connection with G-optimal design

We briefly highlight the connection with G-optimal design. G-optimal design would choose a design matrix $\Sigma$ such that $\|\overline{\phi}_{\pi,p}\|_{\Sigma^{-1}}$ is as small as possible for all possible $\pi$. Since we cannot choose the features in the online setting, a first relaxation is to instead compute (and run) the policy $\pi$ that maximizes the program $\mathcal{U}_{pi}^{\star}(\sigma)$ in every episode $i$. Intuitively, as the area of maximum uncertainty is reached, information is acquired there and the uncertainty is progressively reduced, even though this might be not the most efficient way to proceed from an information-theoretic standpoint. Such procedure would operate in an online fashion, but unfortunately it requires an intractable optimization in policy space. Nonetheless this is the first relaxation to G-optimal design. To obtain the second relaxation, it is useful to consider the alternative definition $\mathcal{U}_{pi}^{\star}(\sigma) = \max_{\pi, \|\theta^{\mathcal{U}}\|_{\Sigma_{pi}} \leq \sqrt{\sigma}} \overline{\phi}_{\pi,p}^{\top}\theta^{\mathcal{U}}$. If we relax the constraint $\|\theta^{\mathcal{U}}\|_{\Sigma_{pi}} \leq \sqrt{\sigma}$ to obtain $\|\theta^{\mathcal{U}}\|_{\Sigma_{pi}} \lesssim \sqrt{d_p\sigma}$ then the feasible space is large enough that random sampling from the feasible set (and computing the maximizing policy by using LSVI) achieves the goal of overestimating the maximum of the unrelaxed program; in particular, sampling $\xi_{pi} \sim \mathcal{N}(0, \sigma\Sigma_{pi}^{-1})$ satisfies the relaxed constraints with high probability and is roughly uniformly distributed in the constraint set.

## 6 Discussion

This works makes progress in relaxing the optimistic closure assumptions on the function class for exploration through a statistically and computationally efficient PAC algorithm. From an algorithmic standpoint, our algorithm is inspired by [Osband et al., 2016b], but from an analytical standpoint, it is justified by a design-of-experiments approach [Lattimore and Szepesvari, 2020]. Remarkably, our approximations to make G-experimental design implementable *online* and with polynomial *computational complexity* only add a $d$ factor compared to G-optimal design. The proof technique is new to our knowledge both in principles and in execution, and can be appreciated in the appendix. We hope that the basic principle is general enough to serve as a foundation to develop new algorithms with even more general function approximators. The contribution to reward-free exploration [Jin et al., 2020a] to linear value functions is also a contribution to the field.

## 7 Broader Impact

This work is of theoretical nature and aims at improving our core understanding of reinforcement learning; no immediate societal consequences are anticipated as a result of this study.

## Acknowledgment

Funding in direct support of this work: Total Innovation Program Fellowship, ONR YIP and NSF career. The authors are grateful to the reviewers for their useful comments, in particular about the explorability requirement.

## Footnotes

[1]This is an approximation to $G$-optimal design, because $\pi$ here is the policy that leads to the most uncertain direction $\bar{\phi}_\pi$ rather than to the direction that reduces the uncertainty the most.

[2] FRANCIS requires only polynomial calls to LSVI and samples from a multivariate normal, see appendix D.7.

[3] G-optimal design does this optimally, but requires choosing the features, which is only possible if one has access to a generative model or in a bandit problem.

[7]For infinite horizon MDPs, these normally coincide.

[8]Note that if $\widehat{V}_{t+1} \in R \times \mathcal{V}_{t+1}$ (the set $\mathcal{V}_{t+1}$ where all elements are scaled by the scalar $R$) then the bounds still hold provided that they are rescaled by $R$.

[9]notice that we are not accounting for the the progress made in episodes where $\mathcal{E}_k$ does not occur

[10]This condition is recurrent in this proof, and is used to invoke lemma 14 *(Derandomization)*, but if it doesn't hold the thesis is automatically satisfied.

[11]Both assumptions are satisfied by the assumptions of the main theorem.

[12]We sometime say we are outside of the failure event to mean we are in the good event for FRANCIS, see definition 7 *(Good Event for* FRANCIS*)*. In particular, the computation in lemma 19 *(Probability of Good Event for* FRANCIS*)* together with the proof in lemma 18 *(Learning a Level)* would provide values for $\delta''$ and for the constants $c_e, c_\alpha, c_\sigma$ if carried out explicitly.

[13]Some symbols, like $i_{max}, k_{max}$ are defined directly in the lemma where the bound is used.

[14]See for example exercise 2.5 in Wainwright [2019].

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

# Contents

# A Preliminaries

## A.1 Symbols

Table 2: Symbols

| | | |
|---|---|---|
| $r_t(s,a)$ | $\stackrel{def}{=}$ | expected reward in $(s,a,t)$ |
| $p_t(s,a)$ | $\stackrel{def}{=}$ | transition function in $(s,a,t)$ |
| $s_{tk}$ | $\stackrel{def}{=}$ | experienced state at timestep $t$ in episode $k$ in phase $t$ |
| $a_{tk}$ | $\stackrel{def}{=}$ | experienced action at timestep $t$ in episode $k$ in phase $t$ |
| $r_{tk}$ | $\stackrel{def}{=}$ | experienced reward[4] at timestep $t$ in episode $k$ in phase $t$ |
| $s_{t+1,k}^{+}$ | $\stackrel{def}{=}$ | experienced state at timestep $t+1$ in episode $k$ in phase $t$ |
| $L_\phi$ | $\stackrel{def}{=}$ | upper bound on $\sup_{s,a,t}\|\phi_t(s,a)\|_2$ |
| $\phi_{tk}$ | $\stackrel{def}{=}$ | $\phi_t(s_{tk},a_{tk})$ |
| $\Sigma_{tk}$ | $\stackrel{def}{=}$ | $\sum_{i=1}^{k-1}\phi_{ti}\phi_{ti}^\top$ |
| $\Sigma_t$ | $\stackrel{def}{=}$ | $\Sigma_{tk}$ matrix after FRANCIS has completed learning in phase $t$ ($k$ is the last episode in that phase) |
| $\mathcal{T}_t(Q_{t+1})(s,a)$ | $\stackrel{def}{=}$ | $r_t(s,a) + \mathbb{E}_{s'\sim p_t(s,a)}Q_{t+1}(s,a)$ |
| $\mathcal{T}_t^P(Q_{t+1})(s,a)$ | $\stackrel{def}{=}$ | $\mathbb{E}_{s'\sim p_t(s,a)}Q_{t+1}(s,a)$ |
| $\mathring{\theta}_t(Q_{t+1})$ | $\stackrel{def}{=}$ | any $\mathring{\theta}_t(Q_{t+1}) \in \mathcal{B}_t$ s.t. $\max_{(s,a)}\big|[\phi_t(s,a)^\top\mathring{\theta}_t(Q_{t+1}) - \mathcal{T}_t^P(Q_{t+1})(s,a)]\big| \le \mathcal{I}(\mathcal{Q}_t,\mathcal{Q}_{t+1})$ when $Q_{t+1} \in \mathcal{Q}_{t+1}$ |
| $\Delta_{ti}(Q_{t+1})$ | $\stackrel{def}{=}$ | $\mathring{Q}_t(Q_{t+1})(s_{ti},\pi_{ti}(s_{ti})) - \mathcal{T}_t^P(Q_{t+1})(s_{ti},\pi_{ti}(s_{ti}))$ |
| $\widehat{\theta}_t$ | $\stackrel{def}{=}$ | $\Sigma_t^{-1}\sum_{i=1}^{n(t)}\phi_{ti}\big[\widehat{V}_{t+1}(s_{t+1,i}^+)\big]$ |
| $\pi_{ti}$ | $\stackrel{def}{=}$ | policy played in episode $i$ of phase $t$ |
| $Q_t(\theta)$ | $\stackrel{def}{=}$ | action value function $(s,a) \mapsto \phi_t(s,a)^\top\theta$ |
| $V_t(\theta)$ | $\stackrel{def}{=}$ | value function $s \mapsto \max_a \phi_t(s,a)^\top\theta$ |
| $\eta_{ti}^t(\widehat{V}_{t+1})$ | $\stackrel{def}{=}$ | $\widehat{V}_{t+1}(s_{t+1,i}^+) - \mathbb{E}_{s'\sim p(s_{ti},\pi_{ti}(s_{ti}))}\widehat{V}_{t+1}(s')$ |
| $\Delta_t^r(s,a)$ | $\stackrel{def}{=}$ | $r_t(s,a) - \phi_t(s,a)^\top\theta_t^r$ |
| $\Delta_{ti}^r$ | $\stackrel{def}{=}$ | $r_t(s_{ti},a_{ti}) - \phi_{ti}^\top\theta_t^r$ |
| $\eta_{ti}^r$ | $\stackrel{def}{=}$ | $r_{ti} - r_t(s_{ti},a_{ti})$ (reward noise) |
| $\mathcal{I}(\mathcal{Q}_t,\mathcal{Q}_{t+1})$ | $\stackrel{def}{=}$ | $\max_{Q_{t+1}\in\mathcal{Q}_{t+1}}\min_{Q_t\in\mathcal{Q}_t}\max_{(s,a)}\big|[Q_t - \mathcal{T}_t^P(Q_{t+1})](s,a)\big|$ |
| $E_t$ | $\stackrel{def}{=}$ | approximation error for the reward, see eq. (157) |
| $k$ | $\stackrel{def}{=}$ | is an overestimate[5] of the number of episodes and is used in the definition of the $\beta$'s below |
| $\delta'$ | $\stackrel{def}{=}$ | is[6] used in the definition of the $\beta$'s below |
| $\beta_t^t$ | $\stackrel{def}{=}$ | $\sqrt{2}\times 2\sqrt{\frac{d_t}{2}\ln\left(1+L_\phi^2 k/d_t\right) + d_{t+1}\ln(1+4\mathcal{R}_{t+1}/(2L_\phi\sqrt{k})) + \ln\left(\frac{1}{\delta'}\right)} + 2$ |
| $\beta_t^r$ | $\stackrel{def}{=}$ | $\sqrt{d_t\ln\left(\frac{1+kL_\phi^2}{\delta'}\right)} + \|\theta_t^r\|_2$ |
| $D_p$ | $\stackrel{def}{=}$ | $d_p\ln(1+kL_\phi^2/d_p)$ |
| $\beta_t^E$ | $\stackrel{def}{=}$ | $\beta_t^r$ |
| $\sqrt{\alpha_t}$ | $\stackrel{def}{=}$ | $3\left(\sqrt{\beta_t^t} + \sqrt{\beta_t^r} + 2\right) = \widetilde{O}(\sqrt{d_t+d_{t+1}})$ |

[4]this only applies if the reward function is learned from data; since we're doing reward free exploration, it instead represents the reward used to populate the dataset $\mathcal{D}$ after FRANCIS has terminated.

[5]in particular it can be set to be equal to $n(t)$ and is $poly(d_1,\cdots,d_H,H,\frac{1}{\epsilon},\frac{1}{\delta})$

[6]in particular it is $\frac{\delta}{poly(d_1,\cdots,d_H,H,\frac{1}{\epsilon},\frac{1}{\delta})}$

$$n(t) \stackrel{def}{=} \text{number of samples collected in phase } t$$

$$\widehat{\theta}_t \stackrel{def}{=} \Sigma_t^{-1} \sum_{i=1}^{n(t)} \phi_{ti} \left[ \widehat{V}_{t+1}(s_{t+1,i}^+) \right]$$

$$\widehat{\theta}_t^r \stackrel{def}{=} \Sigma_t^{-1} \sum_{i=1}^{n(t)} \phi_{ti} \left[ r_{tk} \right]$$

$$\widehat{\theta}_t^{R+PV} \stackrel{def}{=} \widehat{\theta}_t^r + \widehat{\theta}_t$$

$$\mathcal{R}_t \stackrel{def}{=} \text{radius at timestep } t \text{ (but these will be all equal to 1 in the end)}$$

$$\mathcal{R} \stackrel{def}{=} \mathcal{R}_1 = \cdots = \mathcal{R}_H = 1$$

$$q \stackrel{def}{=} \Phi(-3) \text{ (normal cdf evaluated at } -3)$$

$$\mathcal{C}_k \stackrel{def}{=} \left\{ \max_{\pi,\eta \in \mathbb{R}^{d_p} : \|\eta\|_{\Sigma_{pk}} \le \sqrt{\sigma}} \overline{\phi}_{\pi,p}^\top \eta > \epsilon'' > \overline{\epsilon} \right\}$$

$$\mathcal{E}_k \stackrel{def}{=} \left\{ \mathbb{E}_{x_1 \sim \rho} \widehat{V}_{1k}(x_1) - \overline{\epsilon} \ge \max_{\pi,\eta \in \mathbb{R}^{d_p} : \|\eta\|_{\Sigma_{pk}} \le \sqrt{\sigma}} \overline{\phi}_{\pi,p}^\top \eta \right\}$$

$$k(e,i) \stackrel{def}{=} \text{episode in epoch } e \text{ (of a certain phase) such that } E_k \text{ happens for the } i\text{-th time.}$$

$$\zeta_{pk(e,i)} \stackrel{def}{=} \overline{\phi}_{\pi_{k(e,i)},p}^\top \xi_{p,k(e,i)} - \phi_{p,k(e,i)}^\top \xi_{p,k(e,i)}$$

$$A \stackrel{def}{=} \sqrt{8 \ln(\frac{1}{\delta''})}$$

$$\gamma_t(\sigma) \stackrel{def}{=} \sqrt{2\sigma_t d_t \ln \frac{2d_t}{\delta''}}$$

$$\pi_t(s) \stackrel{def}{=} \text{indicates the action taken at timestep } t \text{ by policy } \pi \text{ in state } s$$

$$\sigma_{Start} \stackrel{def}{=} 1 / \left( 8d_p \ln \frac{2d_p}{\delta''} \right)$$

$$a\mathcal{B}_t \stackrel{def}{=} \{ax \mid x \in \mathcal{B}_t\} \text{ for a positive real } a$$

$$V_t^\pi \stackrel{def}{=} \text{value function of policy } \pi \text{ at timestep } t \text{ on } \mathcal{M} \text{ once the reward function is fixed}$$

$$V^\star \stackrel{def}{=} \text{optimal value function on } \mathcal{M} \text{ once the reward function is fixed}$$

$$\pi^\star \stackrel{def}{=} \text{optimal policy on } \mathcal{M} \text{ once the reward function is fixed}$$

$$c_e, c_\alpha, c_\sigma \stackrel{def}{=} \text{constants implicitly determined, see proof of theorem 1 and footnote in that page}$$

## A.2 Inherent Bellman Error

**Definition 4** (Inherent Bellman Error and Best Approximator). *Given two compact linear functional spaces[7] $\mathcal{Q}_t$ and $\mathcal{Q}_{t+1}$, the inherent Bellman error at step $t$ is the maximum (in absolute value) residual*

$$\mathcal{I}(\mathcal{Q}_t, \mathcal{Q}_{t+1}) \overset{def}{=} \max_{Q_{t+1} \in \mathcal{Q}_{t+1}} \min_{Q_t \in \mathcal{Q}_t} \max_{(s,a)} |[Q_t - \mathcal{T}_t^P(Q_{t+1})](s,a)|.$$

*The approximator $\mathring{Q}_t(Q_{t+1}) \in \mathcal{Q}_t$ of $Q_{t+1} \in \mathcal{Q}_{t+1}$ through $\mathcal{T}_t^P$ is defined by its parameter $\mathring{\theta}_t(Q_{t+1})$ as any solution $\theta_t \in \mathcal{B}_t$ that verifies (this always exists from the above display) for any $Q_{t+1} \in \mathcal{Q}_{t+1}$*

$$\max_{(s,a)} \left| [\phi_t(s,a)^\top \mathring{\theta}_t(Q_{t+1}) - \mathcal{T}_t^P(Q_{t+1})(s,a)] \right| \leq \mathcal{I}(\mathcal{Q}_t, \mathcal{Q}_{t+1}) \tag{6}$$

*The Bellman residual function $\overline{\Delta}_{\pi,t}$ under policy $\pi$ is implicitly defined in the error decomposition below:*

$$\mathcal{T}_t^P(Q_{t+1})(s,a) \overset{def}{=} \mathring{Q}_t(Q_{t+1})(s,a) + \Delta_t(Q_{t+1})(s,a). \tag{7}$$

*and it satisfies*

$$\mathcal{I}(\mathcal{Q}_t, \mathcal{Q}_{t+1}) = \max_{\substack{(s,a) \\ Q_{t+1} \in \mathcal{Q}_{t+1}}} |\Delta_t(Q_{t+1})(s,a)| \tag{8}$$

We briefly argue why we have the last equality in the above definition

$$\mathcal{I}(\mathcal{Q}_t, \mathcal{Q}_{t+1}) \geq \max_{Q_{t+1} \in \mathcal{Q}_{t+1}} \max_{(s,a)} |\mathring{Q}_t(Q_{t+1})(s,a) - \mathcal{T}_t^P(Q_{t+1})(s,a)| \tag{9}$$

$$= \max_{Q_{t+1} \in \mathcal{Q}_{t+1}} \max_{(s,a)} |\Delta_t(Q_{t+1})(s,a)| \tag{10}$$

where the second step uses eq. (7).

We are going to use the following property throughout the appendix:

**Proposition 1** (Positive Homogeneity of Inherent Bellman Error of System Dynamics). *Let $\gamma$ be a positive scalar number. If*

$$\max_{Q_{t+1} \in \mathcal{Q}_{t+1}} \min_{Q_t \in \mathcal{Q}_t} \max_{(s,a)} |[Q_t - \mathcal{T}_t^P(Q_{t+1})](s,a)| \leq \mathcal{I}(\mathcal{Q}_t, \mathcal{Q}_{t+1}) \tag{11}$$

*then*

$$\max_{Q_{t+1} \in \gamma \mathcal{Q}_{t+1}} \min_{Q_t \in \gamma \mathcal{Q}_t} \max_{(s,a)} |[Q_t - \mathcal{T}_t^P(Q_{t+1})](s,a)| \leq \gamma \mathcal{I}(\mathcal{Q}_t, \mathcal{Q}_{t+1}) \tag{12}$$

*where*

$$\gamma \mathcal{Q}_\tau = \{Q_\tau \mid Q_\tau(s,a) = \phi_\tau(s,a)^\top \theta, \|\theta\|_2 \leq \gamma \mathcal{R}_\tau\}, \quad \tau \in \{t, t+1\}. \tag{13}$$

*This implies that if $\|\theta_{t+1}\|_2 \leq \gamma \mathcal{R}_{t+1}$ then we can find a $\mathring{\theta}_t$ satisfying $\|\mathring{\theta}_t(V_{t+1}(\theta_{t+1}))\|_2 \leq \gamma \mathcal{R}_t$.*

*Proof.* Notice that when we write $\max_x f(x) \leq I$ (for a generic scalar function $f$, an element $x$ in a set, and a scalar $I$) we can replace the statement with $\forall x, f(x) \leq I$ and viceversa:

$$\max_x f(x) \leq I \longleftrightarrow \forall x, \ f(x) \leq I \tag{14}$$

Likewise:

$$\max_x \min_y f(x,y) \leq I \longleftrightarrow \forall x, \exists y: \ f(x,y) \leq I \tag{15}$$

We can recast the Bellman error condition as

$$\forall Q_{t+1} \in \mathcal{Q}_{t+1}, \exists Q_t \in \mathcal{Q}_t: \max_{(s,a)} |[Q_t - \mathcal{T}_t^P(Q_{t+1})](s,a)| \leq \mathcal{I}(\mathcal{Q}_t, \mathcal{Q}_{t+1}) \tag{16}$$

Now consider the bijection

$$Q_t \in \mathcal{Q}_t \longleftrightarrow Q_t' = \gamma Q_t \in \gamma \mathcal{Q}_t,$$

$$Q_{t+1} \in \mathcal{Q}_{t+1} \longleftrightarrow Q_{t+1}' = \gamma Q_{t+1} \in \gamma \mathcal{Q}_{t+1},$$

$$\tag{17}$$

We have that the statement below

$$\forall Q'_{t+1} \in \gamma \mathcal{Q}_{t+1}, \; \exists Q'_t \in \gamma Q_t : \max_{(s,a)} |[Q_t - \mathcal{T}_t^P(Q_{t+1})](s,a)| \le \gamma \mathcal{I}(\mathcal{Q}_t, \mathcal{Q}_{t+1}) \tag{18}$$

holds if and only if

$$\forall Q_{t+1} \in \mathcal{Q}_{t+1}, \; \exists Q_t \in \mathcal{Q}_t : \max_{(s,a)} |[\gamma Q_t - \mathcal{T}_t^P(\gamma Q_{t+1})](s,a)| \le \gamma \mathcal{I}(\mathcal{Q}_t, \mathcal{Q}_{t+1}) \tag{19}$$

holds. Therefore, it suffices to prove eq. (19) to prove the statement. Notice that by linearity of expectation for any $\gamma > 0$ we have

$$\mathcal{T}_t^P Q_{t+1}(\gamma \theta_{t+1}))(s,a) = \mathbb{E}_{s' \sim p_t(s,a)} \max_{a'} [\gamma Q_{t+1}(\theta_{t+1})(s',a')]] \tag{20}$$

$$= \gamma \, \mathbb{E}_{s' \sim p_t(s,a)} \max_{a'} [Q_{t+1}(\theta_{t+1})(s',a')] \tag{21}$$

$$= \gamma \mathcal{T}_t^P(Q_{t+1})(\theta_{t+1})(s,a). \tag{22}$$

Therefore

$$\max_{(s,a)} |[\gamma Q_t - \mathcal{T}_t^P(\gamma Q_{t+1})](s,a)| = \gamma \max_{(s,a)} |[Q_t - \mathcal{T}_t^P(Q_{t+1})](s,a)| \tag{23}$$

The hypothesis of the lemma implies

$$\forall Q_{t+1} \in \mathcal{Q}_{t+1}, \; \exists Q_t \in \mathcal{Q}_t : \gamma \max_{(s,a)} |[Q_t - \mathcal{T}_t^P(Q_{t+1})](s,a)| \le \gamma \mathcal{I}(\mathcal{Q}_t, \mathcal{Q}_{t+1}) \tag{24}$$

and the prior display implies that eq. (19) holds, and so does eq. (18) which is equivalent to eq. (12).

Finally to conclude the proof of the theorem notice that if $\theta_{t+1} \in \gamma \mathcal{R}_{t+1}$ then we can find a $\mathring{\theta}_t \in \gamma \mathcal{R}_t$ such that the Bellman error is at most $\gamma \mathcal{I}(\mathcal{Q}_t, \mathcal{Q}_{t+1})$. $\quad\square$

# B  Analysis of vanilla LSVI

We recall the popular LSVI protocol [Munos, 2005, Munos and Szepesvári, 2008] operating on a *batch* dataset $\mathcal{D} = \{ \left( s_{tk}, a_{tk}, r_{tk}, s^+_{t+1,k} \right) \}^{t=1,\ldots,H}_{k=1,\ldots,n(t)}$ of experienced state-action-reward-successor states. We use $n(t)$ to denote the number of samples collected at a certain timestep $t$. The regularization parameter is optional and defaults to $\lambda = 1$. The LSVI algorithm is used without reward from the dataset $\mathcal{D}$ when called by FRANCIS; instead a pseudoreward function $\text{R}_p$ is prescribed in the last timestep.

---

**Algorithm 2** LSVI($H$,$\text{R}_H$, $\mathcal{D}$; $\lambda = 1$) - This is for use in FRANCIS with reward signal $\text{R}_H$

---

1: **Input**: horizon $H$, dataset $\mathcal{D}$, regularization $\lambda$.
2: Extract pseudo-reward parameter $\xi_H$ from $\text{R}_H$ function
3: Set $\widehat{\theta}_H = \xi_H$
4: **for** timestep $t = H - 1, \ldots, 1$ **do**
5:     Solve $\widehat{\theta}_t = \arg\min_\theta \sum_{k=1}^{n(t)} \left[ \phi_t(s_{tk}, a_{tk})^\top \theta - \max_{a'} \phi_{t+1}(s^+_{t+1,k}, a')^\top \widehat{\theta}_{t+1} \right]^2 + \lambda \|\theta\|_2^2$
6: **end for**
7: **Return** $\pi : (s,t) \mapsto \arg\max_a \phi_t(s,a)^\top \widehat{\theta}_t$

---

---

**Algorithm 3** LSVI($H$, $\mathcal{D}$; $\lambda = 1$) - This is the regular batch algorithm

---

1: **Input**: horizon $H$, dataset $\mathcal{D}$, regularization $\lambda$.
2: **Set** $\widehat{\theta}^{R+PV}_{H+1} = 0$.
3: **for** timestep $t = H, H - 1, \ldots, 1$ **do**
4:     Solve $\widehat{\theta}^{R+PV}_t = \arg\min_\theta \sum_{k=1}^{n(t)} \left[ \phi_t(s_{tk}, a_{tk})^\top \theta - r_{tk} - \max_{a'} \phi_{t+1}(s^+_{t+1,k}, a')^\top \widehat{\theta}^{R+PV}_{t+1} \right]^2 + \lambda \|\theta\|_2^2$
5: **end for**
6: **Return** $\pi : (s,t) \mapsto \arg\max_a \phi_t(s,a)^\top \widehat{\theta}^{R+PV}_t$

---

## B.1  Single Step Error Decomposition

**Lemma 7** (Analysis of Transition Error in Parameter Space). *Let $n(t)$ be the number of episodes where samples have been collected at timestep $t$. If $\widehat{\theta}_t$ satisfies*

$$\widehat{\theta}_t = \Sigma_t^{-1} \sum_{i=1}^{n(t)} \phi_{ti} \left[ \widehat{V}_{t+1}(s^+_{t+1,i}) \right] \tag{25}$$

*then it must also satisfy:*

$$\widehat{\theta}_t = \mathring{\theta}_t(\widehat{V}_{t+1}) + \Sigma_t^{-1} \left( \sum_{i=1}^{n(t)} \phi_{ti} \left[ \Delta_{ti}(\widehat{V}_{t+1}) + \eta^t_{ti}(\widehat{V}_{t+1}) \right] - \lambda \mathring{\theta}_t(\widehat{V}_{t+1}) \right). \tag{26}$$

*Proof.* Let $\pi_{ti}$ be the policy used to generate the rollouts of episode $i$ of phase $t$. Define the trajectory noise of episode $i$ of phase $t$ using the next-state value function $\widehat{V}_{t+1}$ as:

$$\eta^t_{ti}(\widehat{V}_{t+1}) \overset{def}{=} \widehat{V}_{t+1}(s^+_{t+1,i}) - \mathbb{E}_{s' \sim p(s_{ti}, \pi_{ti}(s_{ti}))} \widehat{V}_{t+1}(s'). \tag{27}$$

From eq. (25) we can rewrite the unique solution for $\widehat{\theta}_t$ as

$$\widehat{\theta}_t = \Sigma_t^{-1} \sum_{i=1}^{n(t)} \phi_{ti} \left[ \mathbb{E}_{s' \sim p(s_{ti}, \pi_{ti}(s_{ti}))} \widehat{V}_{t+1}(s') + \eta^t_{ti}(\widehat{V}_{t+1}) \right] \tag{28}$$

Recall the error decomposition of eq. (7) with $(s,a) = (s_{ti}, \pi_{ti}(s_{ti}))$, $\phi_{ti} = \phi(s,a)$, $\Delta_{ti} = \Delta_t(s,a)$

$$\mathbb{E}_{s' \sim p(s,a)} \widehat{V}_{t+1}(s') = \phi_{ti}^\top \mathring{\theta}_t(Q_{t+1}) + \Delta_{ti}(Q_{t+1}) \tag{29}$$

where $\mathring{\theta}_t(Q_{t+1}) \in \mathcal{B}_t$.

Plugging back eq. (29) into eq. (28) gives:

$$\widehat{\theta}_t = \Sigma_t^{-1} \left( \sum_{i=1}^{n(t)} \phi_{ti} \left[ \phi_{ti}^\top \mathring{\theta}_t(\widehat{V}_{t+1}) + \Delta_{ti}(\widehat{V}_{t+1}) + \eta_{ti}^t(\widehat{V}_{t+1}) \right] + \overbrace{\lambda \mathring{\theta}_t(\widehat{V}_{t+1}) - \lambda \mathring{\theta}_t(\widehat{V}_{t+1})}^{=0} \right) \tag{30}$$

$$= \Sigma_t^{-1} \Sigma_t \mathring{\theta}_t(\widehat{V}_{t+1}) + \Sigma_t^{-1} \left( \sum_{i=1}^{n(t)} \phi_{ti} \left[ \Delta_{ti}(\widehat{V}_{t+1}) + \eta_{ti}^t(\widehat{V}_{t+1}) \right] - \lambda \mathring{\theta}_t(\widehat{V}_{t+1}) \right) \tag{31}$$

$$= \mathring{\theta}_t(\widehat{V}_{t+1}) + \Sigma_t^{-1} \left( \sum_{i=1}^{n(t)} \phi_{ti} \left[ \Delta_{ti}(\widehat{V}_{t+1}) + \eta_{ti}^t(\widehat{V}_{t+1}) \right] - \lambda \mathring{\theta}_t(\widehat{V}_{t+1}) \right) . \tag{32}$$

This proves the lemma. $\qquad\square$

**Lemma 8** (Analysis of Reward Error in Parameter Space). *Let $n(t)$ be the number of episodes where samples have been collected at timestep $t$. If $\widehat{\theta}_t^r$ satisfies*

$$\widehat{\theta}_t^r = \Sigma_t^{-1} \sum_{i=1}^{n(t)} \phi_{ti} r_{tk} \tag{33}$$

*then it must also satisfy:*

$$\widehat{\theta}_t^r = \theta_t^r + \Sigma_t^{-1} \left( \sum_{i=1}^{n(t)} \phi_{ti} \left[ \eta_{ti}^r + \Delta_{ti}^r \right] - \lambda \theta_t^r \right) \tag{34}$$

*Proof.* Let $\pi_{ti}$ be the policy used to generate the rollouts of episode $i$ of phase $t$.

From eq. (33) we can rewrite the unique solution for $\widehat{\theta}_t^r$ as (for the definitions of the symbols see table 2)

$$\widehat{\theta}_t^r = \Sigma_t^{-1} \sum_{i=1}^{n(t)} \phi_{ti} \left[ r_t(s_{ti}, a_{ti}) + \eta_{ti}^r \right]$$

$$= \Sigma_t^{-1} \left( \sum_{i=1}^{n(t)} \phi_{ti} \left[ \phi_{ti}^{\top} \theta_t^r + \Delta_{ti}^r + \eta_{ti}^r \right] + \lambda \theta_t^r - \lambda \theta_t^r \right)$$

$$= \theta_t^r + \Sigma_t^{-1} \left( \sum_{i=1}^{n(t)} \phi_{ti} \left[ \eta_{ti}^r + \Delta_{ti} \right] - \lambda \theta_t^r \right) \tag{35}$$

$\square$

## B.2 Single Step Error Bounds

**Definition 5** (Good Event for LSVI). *Assume $\sqrt{n(t)}\mathcal{I}(\mathcal{Q}_t, \mathcal{Q}_{t+1}) \leq \sqrt{\alpha_t}/3$ and $\sqrt{n(t)}E_t \leq \sqrt{\alpha_t}/3$. We say that LSVI (algorithms 2 and 3) is in the good event when the following bound holds for all $t \in [H]$ with[8] $\widehat{V}_{t+1} \in \mathcal{V}_{t+1}$. The definition of the symbols are reported in table 2:*

$$\|\sum_{i=1}^{n(t)} \phi_{ti}\Delta_{ti}(\widehat{V}_{t+1})\|_{\Sigma_t^{-1}} \leq \sqrt{n(t)}\mathcal{I}(\mathcal{Q}_t, \mathcal{Q}_{t+1}) \tag{36}$$

$$\|\sum_{i=1}^{n(t)} \phi_{ti}\eta_{ti}^t(\widehat{V}_{t+1})\|_{\Sigma_t^{-1}} \leq \sqrt{\beta_t^t} \tag{37}$$

$$\lambda\|\mathring{\theta}_t(\widehat{V}_{t+1})\|_{\Sigma_t^{-1}} \leq \sqrt{\lambda}\mathcal{R}_t \tag{38}$$

$$\|\sum_{i=1}^{n(t)} \phi_{ti}\Delta_{ti}^r\|_{\Sigma_t^{-1}} \leq \sqrt{n(t)}E_t \tag{39}$$

$$\|\sum_{i=1}^{n(t)} \phi_{ti}\eta_{ti}^r\|_{\Sigma_t^{-1}} \leq \sqrt{\beta_t^r} \tag{40}$$

$$\lambda\|\theta_t^r\|_{\Sigma_t^{-1}} \leq \sqrt{\lambda}\|\theta_t^r\|_2. \tag{41}$$

*In addition, the above expressions with the relations in lemma 7 (Analysis of Transition Error in Parameter Space) and lemma 8 (Analysis of Reward Error in Parameter Space) imply:*

$$\begin{aligned}
&\|\widehat{\theta}_t^r - \theta_t^r\|_{\Sigma_t} + \|\widehat{\theta}_t - \mathring{\theta}_t(\widehat{V}_{t+1})\|_{\Sigma_t} \\
&\leq \sqrt{n(t)}\mathcal{I}(\mathcal{Q}_t, \mathcal{Q}_{t+1}) + \sqrt{n(t)}E_t + \sqrt{\beta_t^r} + \sqrt{\beta_t^t} + \sqrt{\lambda}\mathcal{R}_t + \sqrt{\lambda}\|\theta_t^r\|_2 \\
&\leq \sqrt{\alpha_t}
\end{aligned} \tag{42}$$

**Lemma 9** (Probability of Good Event for LSVI). *There exists a parameter $\delta' = \frac{\delta}{poly(d_1,...,d_H,H,\frac{1}{\epsilon})}$, such that the good event of definition 5 holds with probability at least $1 - \delta/2$.*

*Proof.* Since $|\Delta_{ti}(\widehat{V}_{t+1})| \leq \mathcal{I}(\mathcal{Q}_t, \mathcal{Q}_{t+1})$, the projection bound (lemma 8 in [Zanette et al., 2020b]) gives the first inequality in the statement of the theorem. The second inequality is proved in lemma 21 *(Transition Noise High Probability Bound)* respectively. The third inequality follows from lemma 25 *(Change of $\Sigma$-Norm)*. Since $|\Delta_{ti}^r| \leq E_t$ the projection bound (lemma 8 in [Zanette et al., 2020b]) again gives the fourth inequality. The fifth inequality follows from theorem 2 in [Abbasi-Yadkori et al., 2011] with 1-sub-Gaussian noise and the last inequality again follows from lemma 25 *(Change of $\Sigma$-Norm)*. In particular it is possible to choose $\delta'$ (in the definition of the $\beta$'s) such that these statements jointly hold with probability at least $1 - \delta/2$ after a union bound over each statement and the timestep $H$. At this point the statement in eq. (42) follows deterministically by chaining with lemmas 7 and 8. $\square$

## B.3 Iterate Boundness

In this section we discuss the *boundness* in the value function parameter.

**Lemma 10** (Boundness at Intermediate Timesteps for algorithm 2)**.** *On the good event for* LSVI *of definition 5 if*

$$\lambda_{min}(\Sigma_t) \geq 4H^2\alpha_t, \quad \forall t \in [p-1] \tag{43}$$

$$\|\xi_p\|_2 \leq \frac{1}{2} \tag{44}$$

*then*

$$\|\widehat{\theta}_t\|_2 \leq 1, \quad \forall t \in [p]. \tag{45}$$

*Proof.* We proceed by induction, showing that $\widehat{\theta}_t$ due to errors can live in bigger and bigger balls, with radius starting from $\frac{1}{2}$ at timestep $p$ to radius 1 at timestep 1.

**Inductive Hypothesis 2.** $\|\widehat{\theta}_t\|_2 \leq (1 - \frac{t-1}{2H})$.

The inductive statement clearly holds at $t = p$ by hypothesis of the lemma; therefore we focus on the inductive step (notice that the induction goes from $t = p$ down to $t = 1$, so the inductive step assumes the inductive hypothesis holds when written for $t + 1$.)

The inherent Bellman error definition (definition 4 *(Inherent Bellman Error and Best Approximator)*) and proposition 1 *(Positive Homogeneity of Inherent Bellman Error of System Dynamics)* ensures

$$\|\widehat{\theta}_{t+1}\|_2 \leq \left(1 - \frac{t}{2H}\right) \longrightarrow \|\mathring{\theta}_t(V_{t+1}(\widehat{\theta}_{t+1}))\|_2 \leq \left(1 - \frac{t}{2H}\right) \tag{46}$$

In particular, the left statement is ensured by the inductive hypothesis for $t + 1$. Next, under the good event of definition 5 *(Good Event for* LSVI*)*, we have that lemma 25 *(Change of $\Sigma$-Norm)* ensures (writing $\mathring{\theta}_t = \mathring{\theta}_t(V_{t+1}(\widehat{\theta}_{t+1}))$ for short)

$$\sqrt{\alpha_t} \geq \|\widehat{\theta}_t - \mathring{\theta}_t\|_{\Sigma_t} \geq \sqrt{\lambda_{min}(\Sigma_t)}\|\widehat{\theta}_t - \mathring{\theta}_t\|_2 \tag{47}$$

Solving for $\|\widehat{\theta}_t - \mathring{\theta}_t\|_2$ and using the lemma's hypothesis gives

$$\|\widehat{\theta}_t - \mathring{\theta}_t\|_2 \leq \frac{\sqrt{\alpha_t}}{2H\sqrt{\alpha_t}} = \frac{1}{2H}. \tag{48}$$

Combined with the prior display, we deduce

$$\|\widehat{\theta}_t\|_2 \leq \|\widehat{\theta}_t - \mathring{\theta}_t\|_2 + \|\mathring{\theta}_t\|_2 \leq 1 - \frac{t}{2H} + \frac{1}{2H} = 1 - \frac{t-1}{2H}. \tag{49}$$

This shows the inductive step. $\qquad\square$

**Lemma 11** (Boundness at Intermediate Timesteps for algorithm 3). *Under the good event definition 5, fix a positive scalar $R$; if*

$$\lambda_{min}(\Sigma_t) \geq 4H^2\alpha_t, \quad \forall t \in [H] \tag{50}$$

$$\|\theta_t^r\|_2 \leq \frac{R}{H} \tag{51}$$

*then*

$$\|\widehat{\theta}_t^{R+PV}\|_2 = \|\widehat{\theta}_t^R + \widehat{\theta}_t\|_2 \leq 2R, \quad \forall t \in [H]. \tag{52}$$

*Proof.* We proceed by induction, showing that $\widehat{\theta}_t^{R+PV}$ due to errors can live in bigger and bigger balls

**Inductive Hypothesis 3.** $\|\widehat{\theta}_t^{R+PV}\|_2 \leq 2(1 - \frac{t-1}{H})R.$

The inductive statement clearly holds at $t = H + 1$; therefore we focus on the inductive step (notice that the induction goes from $t = H + 1$ down to $t = 1$, so the inductive step assumes the inductive hypothesis holds when written for $t + 1$).

The inherent Bellman error definition (definition 4 *(Inherent Bellman Error and Best Approximator)*) and proposition 1 *(Positive Homogeneity of Inherent Bellman Error of System Dynamics)* ensures

$$\|\widehat{\theta}_{t+1}^{R+PV}\|_2 \leq 2\left(1 - \frac{t}{H}\right)R \longrightarrow \|\mathring{\theta}_t(V_{t+1}(\widehat{\theta}_{t+1}^{R+PV}))\|_2 \leq 2\left(1 - \frac{t}{H}\right)R \tag{53}$$

In particular, the left statement is ensured by the inductive hypothesis for $t + 1$. Next, under the good event of definition 5 *(Good Event for LSVI)* (with a scaling argument by $R$ on the $\|\cdot\|_2$ norm of the regressed parameter) we have that lemma 25 *(Change of $\Sigma$-Norm)* ensures (writing $\mathring{\theta}_t = \mathring{\theta}_t(V_{t+1}(\widehat{\theta}_{t+1}^{R+PV}))$ for short)

$$R\sqrt{\alpha_t} \geq \left(\|\widehat{\theta}_t^r - \theta_t^r\|_{\Sigma_t} + \|\widehat{\theta}_t - \mathring{\theta}_t\|_{\Sigma_t}\right) \geq \sqrt{\lambda_{min}(\Sigma_t)}\left(\|\widehat{\theta}_t^r - \theta_t^r\|_2 + \|\widehat{\theta}_t - \mathring{\theta}_t\|_2\right) \tag{54}$$

Solving for $\left(\|\widehat{\theta}_t^r - \theta_t^r\|_2 + \|\widehat{\theta}_t - \mathring{\theta}_t\|_2\right)$ and using the lemma's hypothesis gives

$$\left(\|\widehat{\theta}_t^r - \theta_t^r\|_2 + \|\widehat{\theta}_t - \mathring{\theta}_t\|_2\right) \leq \frac{\sqrt{\alpha_t}}{2H\sqrt{\alpha_t}}R \leq \frac{R}{2H}. \tag{55}$$

Combined with the prior display, we deduce

$$\|\widehat{\theta}_t^{R+PV}\|_2 \leq \|\widehat{\theta}_t^r - \theta_t^r\|_2 + \|\widehat{\theta}_t - \mathring{\theta}_t\|_2 + \|\theta_t^r\|_2 + \|\mathring{\theta}_t\|_2 \leq \frac{R}{2H} + \frac{R}{H} + 2(1 - \frac{t}{H})R \leq 2(1 - \frac{t-1}{H})R. \tag{56}$$

This shows the inductive step. $\square$

## B.4 Multi-Step Analysis: Error Bounds for LSVI

**Lemma 12** (Telescopic Expansion). *Under the good event of definition 5 for algorithm 2 if*

$$\|\xi_p\|_2 \leq \frac{1}{2} \tag{57}$$

*then the learned parameter*

$$\|\widehat{\theta}_t\|_2 \leq 1, \quad t \in [p]. \tag{58}$$

*Furthermore, for any policy $\pi$*

$$\mathbb{E}_{x_1 \sim \rho} \widehat{Q}_1(x_1, \pi_1(x_1)) \geq -\sum_{t=1}^{p-1} \left[ \mathcal{I}(\mathcal{Q}_t, \mathcal{Q}_{t+1}) + \sqrt{\alpha_t} \|\overline{\phi}_{\pi,t}\|_{\Sigma_t^{-1}} \right] + \mathbb{E}_{x_p \sim \pi} \widehat{Q}_p(x_p, \pi_p(x_p)) \tag{59}$$

*and for the greedy policy $\overline{\pi}$ with respect to $\widehat{Q}$, i.e., $\overline{\pi}_t(s) = \arg\max_a \widehat{Q}_t(s, a)$ it additionally holds that*

$$\mathbb{E}_{x_1 \sim \rho} \widehat{V}_1(x_1) \leq \sum_{t=1}^{p-1} \left[ \mathcal{I}(\mathcal{Q}_t, \mathcal{Q}_{t+1}) + \sqrt{\alpha_t} \|\overline{\phi}_{\overline{\pi},t}\|_{\Sigma_t^{-1}} \right] + \mathbb{E}_{x_p \sim \overline{\pi}} \widehat{V}_p(x_p). \tag{60}$$

*Proof.* On the good event for LSVI of definition 5 *(Good Event for LSVI)* the boundness of the iterate $\widehat{\theta}_t$ is given by lemma 10 *(Boundness at Intermediate Timesteps for algorithm 2)*; we can use Cauchy-Schwartz to write:

$$|\overline{\phi}_{\pi,t}^\top \left( \widehat{\theta}_t - \mathring{\theta}_t(\widehat{V}_{t+1}) \right)| \leq \|\overline{\phi}_{\pi,t}\|_{\Sigma_t^{-1}} \|\widehat{\theta}_t - \mathring{\theta}_t(\widehat{V}_{t+1})\|_{\Sigma_t} \leq \sqrt{\alpha_t} \|\overline{\phi}_{\pi,t}\|_{\Sigma_t^{-1}} \tag{61}$$

Using definition 4 *(Inherent Bellman Error and Best Approximator)* we can write:

$$|\overline{\phi}_{\pi,t}^\top \mathring{\theta}_t(\widehat{V}_{t+1}) - \mathbb{E}_{x_t \sim \pi} \mathcal{T}_t^P \widehat{V}_{t+1}(x_t, \pi_t(x_t))| \leq \mathcal{I}(\mathcal{Q}_t, \mathcal{Q}_{t+1}). \tag{62}$$

Combining the two expression gives:

$$| \mathbb{E}_{x_t \sim \pi} \widehat{Q}_t(x_t, \pi_t(x_t)) - \mathbb{E}_{x_{t+1} \sim \pi} \widehat{V}_{t+1}(x_{t+1})| \tag{63}$$

$$= | \mathbb{E}_{x_t \sim \pi} \left[ \widehat{Q}_t(x_t, \pi_t(x_t)) - \mathcal{T}_t^P \widehat{V}_{t+1}(x_t, \pi_t(x_t)) \right] | \tag{64}$$

$$= |\overline{\phi}_{\pi,t}^\top \widehat{\theta}_t - \mathbb{E}_{x_t \sim \pi} \mathcal{T}_t^P (\widehat{V}_{t+1})(x_t, \pi_t(x_t))| \tag{65}$$

$$= |\overline{\phi}_{\pi,t}^\top \widehat{\theta}_t - \overline{\phi}_{\pi,t}^\top \mathring{\theta}_t(\widehat{V}_{t+1}) + \overline{\phi}_{\pi,t}^\top \mathring{\theta}_t(\widehat{V}_{t+1}) - \mathbb{E}_{x_t \sim \pi} \mathcal{T}_t^P (\widehat{V}_{t+1})(x_t, \pi_t(x_t))| \tag{66}$$

$$\leq |\overline{\phi}_{\pi,t}^\top \widehat{\theta}_t - \overline{\phi}_{\pi,t}^\top \mathring{\theta}_t(\widehat{V}_{t+1})| + |\overline{\phi}_{\pi,t}^\top \mathring{\theta}_t(\widehat{V}_{t+1}) - \mathbb{E}_{x_t \sim \pi} \mathcal{T}_t^P (\widehat{V}_{t+1})(x_t, \pi_t(x_t))| \tag{67}$$

$$\leq \sqrt{\alpha_t} \|\overline{\phi}_{\pi,t}\|_{\Sigma_t^{-1}} + \mathcal{I}(\mathcal{Q}_t, \mathcal{Q}_{t+1}). \tag{68}$$

To show the upper bound if $\pi$ is the greedy policy with respect to $\widehat{Q}$ then we can equivalently write $\widehat{V}_t(x_t) = \widehat{Q}_t(x_t, \pi_t(x_t))$

$$| \mathbb{E}_{x_t \sim \pi} \widehat{V}_t(x_t) - \mathbb{E}_{x_{t+1} \sim \pi} \widehat{V}_{t+1}(x_{t+1})| \leq \sqrt{\alpha_t} \|\overline{\phi}_{\pi,t}\|_{\Sigma_t^{-1}} + \mathcal{I}(\mathcal{Q}_t, \mathcal{Q}_{t+1}). \tag{69}$$

Induction now shows the upper bound.

To show the lower bound, for a generic policy $\widehat{V}_t(x_t) \geq \widehat{Q}_t(x_t, \pi_t(x_t))$ and so

$$\mathbb{E}_{x_t \sim \pi} \widehat{Q}_t(x_t, \pi_t(x_t)) \geq -\sqrt{\alpha_t} \|\overline{\phi}_{\pi,t}\|_{\Sigma_t^{-1}} - \mathcal{I}(\mathcal{Q}_t, \mathcal{Q}_{t+1}) + \mathbb{E}_{x_{t+1} \sim \pi} \widehat{V}_{t+1}(x_{t+1}) \tag{70}$$

$$\geq -\sqrt{\alpha_t} \|\overline{\phi}_{\pi,t}\|_{\Sigma_t^{-1}} - \mathcal{I}(\mathcal{Q}_t, \mathcal{Q}_{t+1}) + \widehat{Q}_{t+1}(x_{t+1}, \pi_{t+1}(x_{t+1})). \tag{71}$$

Induction concludes. $\square$

**Proposition 2** (Batch LSVI Guarantees (algorithm 3)). *Under the good event of definition 5* (Good Event for LSVI) *assume that*

$$\forall t \in [H] \qquad \|\theta_t^r\|_2 \leq \frac{R}{H} \tag{72}$$

*If $\widehat{V}$ and $\widehat{\pi}^\star$ are the value function and policy returned by algorithm 3 then*

$$\mathbb{E}_{x_1 \sim \rho} \left( V_1^\star - \widehat{V}_1 \right)(x_1) \leq \sum_{t=1}^{H} \left[ 2E_t + R \left( \mathcal{I}(\mathcal{Q}_t, \mathcal{Q}_{t+1}) + \sqrt{\alpha_t} \|\overline{\phi}_{\pi^\star, t}\|_{\Sigma_t^{-1}} \right) \right]$$

$$\mathbb{E}_{x_1 \sim \rho} \left( \widehat{V}_1 - V_1^{\widehat{\pi}^\star} \right)(x_1) \leq \sum_{t=1}^{H} \left[ 2E_t + R \left( \mathcal{I}(\mathcal{Q}_t, \mathcal{Q}_{t+1}) + \sqrt{\alpha_t} \|\overline{\phi}_{\widehat{\pi}^\star, t}\|_{\Sigma_t^{-1}} \right) \right]. \tag{73}$$

*Proof.* Boundness of the iterates $\|\widehat{\theta}^r + \widehat{\theta}^{R+PV}\|_2$ is ensured by lemma 11 *(Boundness at Intermediate Timesteps for algorithm 3)*. Consider a generic timestep $t$; using the Bellman equation and the fact that $\widehat{V}_t(x_t) \geq \widehat{Q}_t(x_t, \pi_t^\star(x_t))$ gives

$$\mathbb{E}_{x_t \sim \pi^\star} \left( V_t^\star - \widehat{V}_t \right)(x_t) \leq \mathbb{E}_{x_t \sim \pi^\star} r_t(x_t, \pi_t^\star(x_t)) + \mathbb{E}_{x_{t+1} \sim \pi^\star} V_{t+1}^\star(x_{t+1}) - \mathbb{E}_{x_t \sim \pi^\star} \phi_t(x_t, \pi_t^\star(x_t))^\top \left( \widehat{\theta}_t^r + \widehat{\theta}_t \right) \tag{74}$$

$$\leq E_t + \overline{\phi}_{\pi^\star, t}^\top \theta_t^r + \mathbb{E}_{x_{t+1} \sim \pi^\star} V_{t+1}^\star(x_{t+1}) - \mathbb{E}_{x_t \sim \pi^\star} \phi_{\pi^\star, t}^\top \left( \widehat{\theta}_t^r + \widehat{\theta}_t \right) \tag{75}$$

Next, under the good event of definition 5 we can write:

$$\leq 2E_t + \overline{\phi}_{\pi^\star, t}^\top \theta_t^r + \mathbb{E}_{x_{t+1} \sim \pi^\star} V_{t+1}^\star(x_{t+1}) - \overline{\phi}_{\pi^\star, t}^\top \theta_t^r \tag{76}$$

$$- \mathbb{E}_{x_{t+1} \sim \pi^\star} \widehat{V}_{t+1}(x_{t+1}) + R[\mathcal{I}(\mathcal{Q}_t, \mathcal{Q}_{t+1}) + \sqrt{\alpha_t} \|\overline{\phi}_{\pi^\star, t}\|_{\Sigma_t^{-1}}] \tag{77}$$

Induction gives the first statement.

Now again we start with the definition of expected feature and the Bellman equation:

$$\mathbb{E}_{x_t \sim \widehat{\pi}^\star} \left( \widehat{V}_t - V_t^{\widehat{\pi}^\star} \right)(x_t) = \overline{\phi}_{\widehat{\pi}^\star, t}^\top (\widehat{\theta}_t^r + \widehat{\theta}_t) - \mathbb{E}_{x_t \sim \widehat{\pi}^\star} r_t(x_t, \widehat{\pi}_t^\star(x_t)) - \mathbb{E}_{x_{t+1} \sim \widehat{\pi}^\star} V_{t+1}^{\widehat{\pi}^\star}(x_{t+1}) \tag{78}$$

$$\leq \overline{\phi}_{\widehat{\pi}^\star, t}^\top \theta^r + E_t + R[\mathcal{I}(\mathcal{Q}_t, \mathcal{Q}_{t+1}) + \sqrt{\alpha_t} \|\overline{\phi}_{\widehat{\pi}^\star, t}\|_{\Sigma_t^{-1}}] + \tag{79}$$

$$+ \mathbb{E}_{x_{t+1} \sim \widehat{\pi}^\star} \widehat{V}_{t+1}(x_{t+1}) - \overline{\phi}_{\widehat{\pi}^\star, t}^\top \theta_t^r + E_t + \mathbb{E}_{x_{t+1} \sim \widehat{\pi}^\star} V_{t+1}^{\widehat{\pi}^\star}(x_{t+1}). \tag{80}$$

Induction again concludes. $\square$

## C  Design of Experiments

We show that obtaining $\|\overline{\phi}_{\pi,t}\|_{\Sigma_t^{-1}} \leq \frac{\epsilon}{H\sqrt{\alpha_t}} = \epsilon'$ suffices; we assume $\mathcal{I}(\mathcal{Q}_t, \mathcal{Q}_{t+1}) = E_t = 0$ for simplicity as well as $d_1 = \cdots = d_H$. We immediately have that

$$\sum_{t=1}^{H} \sqrt{\alpha_t}\|\overline{\phi}_{\pi,t}\|_{\Sigma_t^{-1}} \leq H \times \sqrt{\alpha_t} \times \frac{\epsilon}{H\sqrt{\alpha_t}} = \epsilon. \tag{81}$$

Thus, summing the two equations in eq. (73) for any linear reward function with $\|\theta_t\|_2 \leq \frac{1}{H}$ ensures an $\epsilon$-optimal policy on that reward function is returned.

The Kiefer-Wolfowitz theorem in Lattimore and Szepesvári [2020] guarantees such reduction in $\|\overline{\phi}_{\pi,t}\|_{\Sigma_t^{-1}}$ using $\widetilde{O}(d^2 + \frac{d}{(\epsilon')^2}) = \widetilde{O}(d^2 + \frac{dH^2\alpha_t}{\epsilon^2})$ samples at every level / timestep if $G$-optimal design is used. After sampling all levels and substituting the value for $\alpha_t$ in table 2 the sample complexity of doing G-optimal design becomes $\widetilde{O}(d^2 + \frac{d^2H^3}{\epsilon^2})$.

Notice that this setting can model MDPs with rewards in $[0, 1/H]$ and value functions in $[0, 1]$; moving to the standard setting with rewards in $[0, 1]$ and value function in $[0, H]$ adds $H^2$ to the sample complexity to obtain an $\epsilon$-optimal policy.

# D  Analysis of FRANCIS

## D.1  Generating Bounded Iterates

The following lemma ensures FRANCIS generates bounded iterates for an appropriate choice of $\sigma$.

**Lemma 13** (Boundness at Exploratory Timestep). *In episode $k$ of phase $p$, if*

$$\lambda_{min}(\Sigma_{pk}) \geq 8d_p \ln \frac{2d_p}{\delta''} \sigma \tag{82}$$

$$\xi_p \sim \mathcal{N}(0, \sigma \Sigma_{pk}^{-1}) \tag{83}$$

*then*

$$\|\xi_p\|_2 \leq \frac{1}{2} \tag{84}$$

*on the good event of definition 7* (Good Event for FRANCIS).

*Proof.* Directly by the choice of $\sigma$ and the definition of good event for FRANCIS (see definition 7 *(Good Event for FRANCIS)*). $\square$

## D.2  Derandomization

The following lemma relates the sampling of the algorithm to a procedure that selects the policy / parameter leading to the area of highest (scaled) uncertainty.

**Lemma 14** (Derandomization). *Outside of the failure event, assume that for any policy $\pi$,*

$$\sum_{t=1}^{p-1} \left[ \mathcal{I}(\mathcal{Q}_t, \mathcal{Q}_{t+1}) + \sqrt{\alpha_t} \|\overline{\phi}_{\pi,t}\|_{\Sigma_t^{-1}} \right] \leq \overline{\epsilon} \tag{85}$$

*for some scalar $\overline{\epsilon} > 0$. Consider sampling*

$$\xi_p \sim \mathcal{N}(0, \sigma \Sigma_{pk}^{-1}), \tag{86}$$

*define $\mathrm{R}_p(s,a) = \phi_p(s,a)^\top \xi_p$ and let $\widehat{V}$ be the value function computed by $\mathrm{LSVI}(p, \mathrm{R}_p \mathcal{D})$ (see algorithm 2). Then for a fixed constant $q \in \mathbb{R}$*

$$\mathbf{P}\left( \mathbb{E}_{x_1 \sim \rho} \widehat{V}_1(x_1) - \overline{\epsilon} > \max_{\pi, \eta \in \mathbb{R}^{d_p} : \|\eta\|_{\Sigma_{pk}} \leq \sqrt{\sigma}} \overline{\phi}_{\pi,p}^\top \eta \right) \geq q. \tag{87}$$

*if*

$$\max_{\pi, \eta \in \mathbb{R}^{d_p} : \|\eta\|_{\Sigma_{pk}} \leq \sqrt{\sigma}} \overline{\phi}_{\pi,p}^\top \eta \geq \overline{\epsilon} \tag{88}$$

$$\|\xi_p\|_2 \leq \frac{1}{2}. \tag{89}$$

*Proof.* Define the maximizer of the "scaled uncertainty" in a generic episode $k$ of phase $p$:

$$\left( \overset{\triangle}{\pi}, \overset{\triangle}{\eta} \right) \overset{def}{=} \underset{\substack{\pi \\ \|\eta\|_{\Sigma_{pk}} \leq \sqrt{\sigma}}}{\arg\max} |\overline{\phi}_{\overset{\triangle}{\pi},p}^\top \eta| \tag{90}$$

as the policy / parameter that maximizes the uncertainty.

Next, let $\overline{\pi}$ be the policy selected by the agent, through LSVI, corresponding to the sampled parameter $\xi_p$ and let $\widehat{Q}, \widehat{V}$ be the (action) value functions. Since $\overline{\pi}$ is the maximizing policy for $\widehat{Q}$, we must have:

$$\mathbb{E}_{x_1 \sim \rho} \widehat{V}_1(x_1) = \mathbb{E}_{x_1 \sim \rho} \widehat{Q}_1(x_1, \overline{\pi}_1(x_1)) \geq \mathbb{E}_{x_1 \sim \rho} \widehat{Q}_1(x_1, \overset{\triangle}{\pi}_1(x_1)). \tag{91}$$

In addition on the good event for LSVI lemma 12 *(Telescopic Expansion)* gives:

$$\mathbb{E}_{x_1 \sim \rho} \widehat{V}_1(x_1) \geq \mathbb{E}_{x_1 \sim \rho} \widehat{Q}_1(x_1, \overset{\triangle}{\pi}_1(x_1)) \geq \sum_{t=1}^{p-1} \left[ -\mathcal{I}(\mathcal{Q}_t, \mathcal{Q}_{t+1}) - \sqrt{\alpha_t} \|\overline{\phi}_{\overset{\triangle}{\pi},t}\|_{\Sigma_t^{-1}} \right] + \underbrace{\mathbb{E}_{x_p \sim \overset{\triangle}{\pi}} \widehat{Q}_p(x_p, \overset{\triangle}{\pi}_p(x_p))}_{(\overline{\phi}_{\overset{\triangle}{\pi},p})^\top \xi_p}. \tag{92}$$

Subtracting $\bar{\epsilon}$ to both sides and using the hypothesis gives

$$\mathbb{E}_{x_1 \sim \rho} \widehat{V}_1(x_1) - \bar{\epsilon} \geq -2\bar{\epsilon} + (\bar{\phi}_{\underset{\bar{\pi},p}{\triangle}})^\top \xi_p. \tag{93}$$

We can now proceed to bound the quantity of interest:

$$\mathbf{P}\left(\mathbb{E}_{x_1 \sim \rho} \widehat{V}_1(x_1) - \bar{\epsilon} \geq (\bar{\phi}_{\underset{\bar{\pi},p}{\triangle}})^\top \overset{\triangle}{\eta}\right) \tag{94}$$

$$\geq \mathbf{P}\left(-2\bar{\epsilon} + \bar{\phi}_{\underset{\bar{\pi},p}{\triangle}}^\top \xi_p \geq (\bar{\phi}_{\underset{\bar{\pi},p}{\triangle}})^\top \overset{\triangle}{\eta}\right) \tag{95}$$

$$= \mathbf{P}\left(\bar{\phi}_{\underset{\bar{\pi},p}{\triangle}}^\top \xi_p \geq \underbrace{2\bar{\epsilon}}_{\text{Error in Propagating the Uncertainty}} + \underbrace{(\bar{\phi}_{\underset{\bar{\pi},p}{\triangle}})^\top \overset{\triangle}{\eta}}_{\text{Uncertainty in the Level to Learn}}\right) \geq q \tag{96}$$

Notice that $\xi_p$ is independent of $\bar{\phi}_{\underset{\bar{\pi}}{\triangle}}$ when conditioned on the $\Sigma_{tk}$. The last step is an application of lemma 15 *(Uncertainty Overestimation)* as long as the condition

$$\bar{\epsilon} \leq \max_{\phi, \|\eta\|_{\Sigma_{pk}} \leq \sqrt{\sigma}} \bar{\phi}_{\bar{\pi},p}^\top \eta \tag{97}$$

is met. $\qquad\qquad\qquad\qquad\qquad\qquad\qquad\qquad\qquad\qquad\qquad\qquad\qquad\qquad\qquad\qquad\qquad\qquad\square$

**Lemma 15** (Uncertainty Overestimation). *Let $\bar\epsilon, \sigma$ be positive scalars, and let $\Sigma$ be an spd matrix and let*

$$\xi \sim \mathcal{N}(0, \sigma\Sigma^{-1}) \tag{98}$$

*be the associated random vectors. For a fixed vector $\phi$ we have that*

$$\mathbf{P}\left(\phi^\top \xi \geq \max_{\phi, \|\eta\|_\Sigma \leq \sqrt{\sigma}} \phi^\top \eta + 2\bar\epsilon\right) \geq \Phi(-3) \overset{def}{=} q \tag{99}$$

*where $\Phi(\cdot)$ is the normal CDF function as long as the condition*

$$\bar\epsilon \leq \max_{\phi, \|\eta\|_\Sigma \leq \sqrt{\sigma}} \phi^\top \eta = \sqrt{\sigma}\|\phi\|_{\Sigma^{-1}} \tag{100}$$

*holds true.*

*Proof.* Before we prove the statement, we notice that the equivalent expression $\max_{\phi, \|\eta\|_\Sigma \leq \sqrt{\sigma}} \phi^\top \eta = \sqrt{\sigma}\|\phi\|_{\Sigma^{-1}}$ can be found in chapter 19 of [Lattimore and Szepesvári, 2020] about the LINUCB algorithm, see also lemma 26 *(Linear Bandit Exploration Bonus)*. For any fixed $\Sigma$, we have that $\xi \sim \mathcal{N}(0, \sigma\Sigma^{-1})$ is independent of $\phi$ by hypothesis, and so the inner product below is normally distributed

$$\phi^\top \xi \sim \mathcal{N}\left(0, \sigma\phi^\top \Sigma^{-1}\phi\right), \tag{101}$$

or equivalently

$$\phi^\top \xi \sim \mathcal{N}\left(0, \sigma\|\phi\|_{\Sigma^{-1}}^2\right). \tag{102}$$

Rescaling by its standard deviation leads to the following definition:

$$X \overset{def}{=} \frac{\phi^\top \xi}{\sqrt{\sigma}\|\phi\|_{\Sigma^{-1}}} \sim \mathcal{N}(0, 1). \tag{103}$$

The step below follows

$$\mathbf{P}\left(\phi^\top \xi \geq \sqrt{\sigma}\|\phi\|_{\Sigma^{-1}} + 2\bar\epsilon\right) = \mathbf{P}\left(X \geq 1 + \frac{2\bar\epsilon}{\sqrt{\sigma}\|\phi\|_{\Sigma^{-1}}}\right). \tag{104}$$

The rhs above is $\geq \Phi(-3)$ as long as

$$\bar\epsilon \leq \sqrt{\sigma}\|\phi\|_{\Sigma^{-1}}. \tag{105}$$

The thesis follows from the definition of the normal CDF. $\square$

## D.3 Learning an Epoch

The following lemma is key to our analysis and shows the number of episodes required to reduce the scaled uncertainty to the minimum allowable ($\approx \bar{\epsilon} > 0$). In an epoch the value for $\sigma$ is fixed.

**Lemma 16** (Learning an Epoch). *Let $\underline{k}$ and $\overline{k}$ be the starting and ending episodes in epoch $e$ of phase $p$. If the following statements hold:*

1. *for any policy $\pi$ it holds that $\sum_{t=1}^{p-1} \left[ \mathcal{I}(\mathcal{Q}_t, \mathcal{Q}_{t+1}) + \sqrt{\alpha_t} \|\overline{\phi}_{\pi,t}\|_{\Sigma_t^{-1}} \right] \leq \bar{\epsilon}$*

2. *$\lambda_{min}(\Sigma_{p\underline{k}}) \geq 8 d_p \ln \frac{2 d_p}{\delta''} \sigma$ (this ensures boundness of $\|\xi_p\|_2$ in lemma 14 (Derandomization))*

3. *$\frac{L_\phi^2}{\lambda} \leq 1$ (always satisfied by our choice $L_\phi = 1$ and $\lambda = 1$)*

4. *$\lambda > 1$ (always satisfied by our choice $\lambda = 1$)*

*then after at most*

$$k_{max} = \overline{k} - \underline{k} = \left\lceil \frac{2}{1-q} \times \frac{(\sqrt{\gamma(\rho) D_p} + A)^2}{(\epsilon'')^2} \right\rceil \tag{106}$$

*episodes we must have*

$$\max_{\pi, \eta \in \mathbb{R}^{d_p} : \|\eta\|_{\Sigma_{p\overline{k}}} \leq \sqrt{\sigma}} \overline{\phi}_{\pi,p}^\top \eta \leq \epsilon'' \tag{107}$$

*on the good event definition 7 (Good Event for FRANCIS) provided that*

$$\epsilon'' \geq \bar{\epsilon}. \tag{108}$$

*Proof.* First notice that if the eigenvalue condition is satisfied for at a given episode $\underline{k}$ then it must be satisfied for all successive episodes $k \geq \underline{k}$ since $\Sigma_{tk} \succeq \Sigma_{t\underline{k}}$. In particular, define the events

$$\mathcal{C}_k \overset{def}{=} \left\{ \max_{\pi, \eta \in \mathbb{R}^{d_p} : \|\eta\|_{\Sigma_{pk}} \leq \sqrt{\sigma}} \overline{\phi}_{\pi,p}^\top \eta > \epsilon'' > \bar{\epsilon} \right\} \tag{109}$$

$$\mathcal{E}_k \overset{def}{=} \left\{ \mathbb{E}_{x_1 \sim \rho} \widehat{V}_{1k}(x_1) - \bar{\epsilon} \geq \max_{\pi, \eta \in \mathbb{R}^{d_p} : \|\eta\|_{\Sigma_{pk}} \leq \sqrt{\sigma}} \overline{\phi}_{\pi,p}^\top \eta \right\}. \tag{110}$$

We examine what happens in those episodes where $\mathcal{E}_k$ occurs (notice that $\mathbf{P}(\mathcal{E}_k \mid \mathcal{C}_k) \geq q$ thanks to lemma 14 *(Derandomization))*.

Let $k(e, i)$ be the $i$-th consecutive episode index in epoch $e$ of phase $p$ such that $\mathcal{E}_{k(e,i)}$ occurs (so in $k(e,1), k(e,2), \ldots$ we have that $\mathcal{E}_{k(e,1)}, \mathcal{E}_{k(e,2)}$ occurs). Since $\|\xi_{pk(e,i)}\|_2 \leq 1/2$ in the good event of definition 7 *(Good Event for FRANCIS)*, we can use lemma 13 *(Boundness at Exploratory Timestep)* and lemma 12 *(Telescopic Expansion)* to write

$$\mathbb{E}_{x_1 \sim \rho} \widehat{V}_{pk(e,i),1}(x_1) - \bar{\epsilon} \leq \phi_{pk(e,i)}^\top \xi_{pk(e,i)} + \zeta_{pk(e,i)}. \tag{111}$$

where

$$\zeta_{pk(e,i)} \overset{def}{=} \overline{\phi}_{\pi_{k(e,i)},p}^\top \xi_{p,k(e,i)} - \phi_{p,k(e,i)}^\top \xi_{p,k(e,i)} \tag{112}$$

Let $i_{max}$ be a fixed positive constant to be determined later. Taking average of the previous display up to $i_{max}$ gives:

$$\frac{1}{i_{max}} \sum_{i=1}^{i_{max}} \mathbb{E}_{x_1 \sim \rho} \widehat{V}_{pk(e,i),1}(x_1) - \bar{\epsilon} \leq \frac{1}{i_{max}} \sum_{i=1}^{i_{max}} \left( \phi_{pk(e,i)}^\top \xi_{pk(e,i)} + \zeta_{pk(e,i)} \right). \tag{113}$$

Under the good event of definition 7 *(Good Event for FRANCIS)* we have

$$\frac{1}{i_{max}} \sum_{i=1}^{i_{max}} \zeta_{pk(e,i)} \leq \frac{A}{\sqrt{i_{max}}} \tag{114}$$

with $A = \widetilde{O}(1)$. For the remaining term, using Cauchy-Schwartz, and the fact that we are on the good event (see definition 7 *(Good Event for* FRANCIS*)*) gives

$$\frac{1}{i_{max}} \sum_{i=1}^{i_{max}} \phi_{pk(e,i)}^{\top} \xi_{pk(e,i)} \leq \frac{1}{i_{max}} \sum_{i=1}^{i_{max}} \|\phi_{pk(e,i)}\|_{\Sigma_{pk(e,i)}^{-1}} \underbrace{\|\xi_{pk(e,i)}\|_{\Sigma_{pk(e,i)}}}_{\sqrt{\gamma_t(\sigma)}} \tag{115}$$

After one more Cauchy-Schwartz we obtain the upper bound below:

$$\leq \frac{\sqrt{\gamma_t(\sigma)}}{i_{max}} \sum_{i=1}^{i_{max}} \|\phi_{pk(e,i)}\|_{\Sigma_{pk(e,i)}^{-1}} \leq \sqrt{\frac{\gamma_t(\sigma)}{i_{max}} \sum_{i=1}^{i_{max}} \|\phi_{pk(e,i)}\|_{\Sigma_{pk(e,i)}^{-1}}^2} . \tag{116}$$

We focus on the sum of squared features; by lemma 25 *(Change of $\Sigma$-Norm)* and the lemma's hypothesis

$$\|\phi_{pk(e,i)}\|_{\Sigma_{pk(e,i)}^{-1}}^2 \leq \frac{1}{\lambda} \|\phi_{pk(e,i)}\|_2^2 \leq \frac{L_\phi^2}{\lambda} \leq 1 \tag{117}$$

and so the sum of squared features becomes[9] (using the elliptic potential lemma, see lemma 11 in [Abbasi-Yadkori et al., 2011]):

$$\sum_{i=1}^{i_{max}} \|\phi_{pk(e,i)}\|_{\Sigma_{pk(e,i)}^{-1}}^2 = \sum_{i=1}^{i_{max}} \min\{1, \|\phi_{pk(e,i)}\|_{\Sigma_{pk(e,i)}^{-1}}^2\} \leq \ln\left(\frac{\det \Sigma_{pk(e,i_{max})}}{\det \Sigma_{p,\underline{k}}}\right) \leq \ln \det \Sigma_{pk(e,i_{max})}. \tag{118}$$

The last step follows because $\Sigma_{p\underline{k}} \succeq \lambda I \succeq I$, an so $\det(\Sigma_{p\underline{k}}) \geq \det I = 1$. Let $D_p = d_p \ln(1 + kL_\phi^2/d) = \widetilde{O}(d_p)$ be an upper bound to $\ln \det \Sigma_{pk(e,i_{max})}$ (see lemma 10 in Abbasi-Yadkori et al. [2011]). We can claim that an upper bound to eq. (113) is

$$\leq \frac{A + \sqrt{\gamma_t(\sigma)D_p}}{\sqrt{i_{max}}}. \tag{119}$$

Since we're summing over episode indexes where $\mathcal{E}_{k(e,i)}$ holds, it follows that

$$\frac{1}{i_{max}} \sum_{i=1}^{i_{max}} \left[ \max_{\pi,\eta \in \mathbb{R}^{d_p} : \|\eta\|_{\Sigma_{pk(e,i)}} \leq \sqrt{\sigma}} \overline{\phi}_{\pi,p}^{\top} \eta \right] \leq \frac{A + \sqrt{\gamma(\sigma)D_p}}{\sqrt{i_{max}}} \tag{120}$$

if each term in the summation in the lhs is $\geq \epsilon''$ (the condition is needed to apply lemma 14 *(Derandomization)*; if it does not hold the lemma's thesis is satisfied). By lemma 17 *(Uncertainty Lemma)*

$$\max_{\pi,\eta \in \mathbb{R}^{d_p} : \|\eta\|_{\Sigma_{p,k(e,i+1)}} \leq \sqrt{\sigma}} \overline{\phi}_{\pi,p}^{\top} \eta \leq \max_{\pi,\eta \in \mathbb{R}^{d_p} : \|\eta\|_{\Sigma_{p,k(e,i)}} \leq \sqrt{\sigma}} \overline{\phi}_{\pi,p}^{\top} \eta \tag{121}$$

Since the terms in the lhs of eq. (120) are strictly decreasing, the last one must be smaller than the average, which implies we must obtain

$$\max_{\pi,\eta \in \mathbb{R}^{d_p} : \|\eta\|_{\Sigma_{pk(e,i_{max})}} \leq \sqrt{\sigma}} \overline{\phi}_{\pi,p}^{\top} \eta \leq \epsilon'' \tag{122}$$

after

$$i_{max} \geq \frac{(\sqrt{\gamma_t(\rho)D_p} + A)^2}{(\epsilon'')^2} \tag{123}$$

episodes provided that[10]

$$\epsilon'' \geq \overline{\epsilon}. \tag{124}$$

We can finally compute how big $k_{max}$ (the total number of episodes in the epoch) needs to be: from definition 7 *(Good Event for* FRANCIS*)* if

$$k_{max} \geq \frac{1}{4} \times \frac{2\ln(\frac{1}{\delta''})}{1 - q} \tag{125}$$

then we can write

$$\frac{i_{max}}{k_{max}} \geq \frac{1-q}{2}. \tag{126}$$

(recall $i_{max}$ is the the number of episodes where $\mathcal{E}_k$ occurs: $i_{max} = \sum_{k=1}^{k_{max}} \mathbb{1}\{\mathcal{E}_k \mid \mathcal{C}_k\}$). Therefore, a total number of episodes

$$k_{max} = \left\lceil \frac{2}{1-q} \times \frac{(\sqrt{\gamma_t(\rho)D_p} + A)^2}{(\epsilon'')^2} \right\rceil \tag{127}$$

suffices (as this automatically satisfies eq. (125)). $\qquad\square$

**Lemma 17** (Uncertainty Lemma). *Let $\overline{k}$ and $k$ be two generic episodes in an epoch $e$ in phase $p$ such that $\overline{k} \geq k$. We have that*

$$\max_{\pi, \eta \in \mathbb{R}^{d_p} : \|\eta\|_{\Sigma_{p\overline{k}}} \leq \sqrt{\sigma}} \overline{\phi}_{\pi,p}^\top \eta \leq \max_{\pi, \eta \in \mathbb{R}^{d_p} : \|\eta\|_{\Sigma_{pk}} \leq \sqrt{\sigma}} \overline{\phi}_{\pi,p}^\top \eta. \tag{128}$$

*In addition, for positive real numbers $\rho_1 \leq \rho_2$ and a generic spd matrix $\Sigma$ we also have*

$$\max_{\pi, \eta \in \mathbb{R}^{d_p} : \|\eta\|_{\Sigma} \leq \sqrt{\rho_1}} \overline{\phi}_{\pi,p}^\top \eta = \sqrt{\frac{\rho_1}{\rho_2}} \max_{\pi, \eta \in \mathbb{R}^{d_p} : \|\eta\|_{\Sigma} \leq \sqrt{\rho_2}} \overline{\phi}_{\pi,p}^\top \eta. \tag{129}$$

*Proof.* Since $\Sigma_{p\overline{k}} \succeq \Sigma_{pk}$ (this notation means $\Sigma_{p\overline{k}}$ is more positive definite than $\Sigma_{pk}$, more precisely $\phi^\top \Sigma_{p\overline{k}} \phi \geq \phi^\top \Sigma_{pk} \phi$ for all $\phi$) we have the set inclusion

$$\{\eta \mid \|\eta\|_{\Sigma_{p\overline{k}}} \leq \sqrt{\sigma}\} \subseteq \{\eta \mid \|\eta\|_{\Sigma_{pk}} \leq \sqrt{\sigma}\} \tag{130}$$

Since we're maximizing over a smaller set, the first result follows.

For the second statement, recall we can rewrite the programs in eq. (129) (see chapter 19 of [Lattimore and Szepesvári, 2020] about LINUCB or equivalently lemma 26 *(Linear Bandit Exploration Bonus)* ); here we identify the feature of an action in LINUCB with $\overline{\phi}_{\pi,p}$) as

$$\max_{\pi} \sqrt{\rho_1} \|\overline{\phi}_{\pi,p}\|_{\Sigma^{-1}} \tag{131}$$

for the lhs and

$$\max_{\pi} \sqrt{\frac{\rho_1}{\rho_2}} \sqrt{\rho_2} \|\overline{\phi}_{\pi,p}\|_{\Sigma^{-1}} \tag{132}$$

for the rhs, showing equality. $\qquad\square$

## D.4 Learning a Phase

In this section we show how FRANCIS learns a phase (i.e., the *dynamics* at a certain *timestep*) and compute the total number of episodes required to do so. This is where the *explorability* condition is used.

**Lemma 18** (Learning a Level). *Consider phase p and let the following hypotheses hold*

1. $\sum_{t=1}^{p-1}\left[\mathcal{I}(\mathcal{Q}_t,\mathcal{Q}_{t+1}) + \sqrt{\alpha_t}\|\overline{\phi}_{\pi,t}\|_{\Sigma_t^{-1}}\right] \leq \overline{\epsilon}$

2. $\left(\frac{\nu}{\epsilon}\right)^2 \geq 2 \times 8d_p \ln \frac{2d_p}{\delta''}$

*Then after at most ($e_{max} = \widetilde{O}(1)$ and $\sigma_{e_{max}}$ are defined in the proof)*

$$n(t) = \left\lceil \frac{2}{1-q} \times \frac{(\sqrt{\gamma_t(\sigma_{e_{max}})D_p} + A)^2}{\epsilon^2} \right\rceil \times e_{max} = \widetilde{O}\left(\frac{d_p^2 H^2 \alpha_p}{\epsilon^2}\right) = \widetilde{O}\left(\frac{d_p^2 \times H^2(d_p + d_{p+1})}{\epsilon^2}\right) \tag{133}$$

*episodes it must hold that*

$$\max_{\pi,\eta\in\mathbb{R}^{d_p}:\|\eta\|_{\Sigma_{p\overline{k}}}\leq\sqrt{\alpha_p}} \overline{\phi}_{\pi,p}^\top\eta \leq \frac{\epsilon}{2H} \tag{134}$$

*Proof.* Let $\sigma_1,\sigma_2,\dots$ be the sequences of the $\sigma$ parameter chosen in the different epochs, and additionally

$$\sigma_{Start} = 1/\left(8d_p \ln \frac{2d_p}{\delta''}\right). \tag{135}$$

We proceed by induction, with the following inductive hypothesis:

**Inductive Hypothesis 4.** *In phase p the following conditions hold*

(a) $\lambda_{min}(\Sigma_{pk(e,1)}) \geq 8d_p \ln \frac{2d_p}{\delta''}\sigma_e$  *(at the beginning of epoch e)*

(b) $\sigma_e = 2^{e-1}\sigma_{Start}$  *(at the beginning of epoch e)*

To show that the inductive hypothesis is satisfied in the base case ($e = 1$), notice that $(b)$ holds by definition and $(a)$ holds by setting $\lambda = 1$. Now we show the inductive step.

Since the inductive hypothesis satisfies the hypothesis of lemma 16 *(Learning an Epoch)*, on the good event definition 7 *(Good Event for FRANCIS)* it immediately follows that

$$\max_{\pi,\eta\in\mathbb{R}^{d_p}:\|\eta\|_{\Sigma_{pk}}\leq\sqrt{\sigma_e}} \overline{\phi}_{\pi,p}^\top\eta \leq \epsilon'' \tag{136}$$

after $k_{max}$ episodes (see lemma 16 *(Learning an Epoch)*). Here in particular $k$ is the last episode of epoch $e$. The explorability condition in definition 2 *(Explorability)* implies that

$$\forall\eta\neq 0,\ \exists\pi \quad \text{such that} \quad \overline{\phi}_{\pi,t}^\top\frac{\eta}{\|\eta\|_2} \geq \nu_{min}. \tag{137}$$

Consider the normalized evector $v$ corresponding to the minimum eigenvalue $q > 0$ for $\Sigma_{pk}$ and define:

$$\eta = qv. \tag{138}$$

We're interested in determining the maximum $q$ so that the constraint in the program eq. (136) is still satisfied, i.e., the condition below

$$\sigma_e \geq \|qv\|_{\Sigma_{pk}}^2 = (qv)^\top \Sigma_{pk}(qv) = q^2\lambda_{min}(\Sigma_{pk}) \tag{139}$$

gives the maximum value for $q$

$$q = \sqrt{\frac{\sigma_e}{\lambda_{min}(\Sigma_{pk})}} \tag{140}$$

in order for $qv$ to satisfy $\|qv\|_{\Sigma_{pk}} \leq \sqrt{\sigma_e}$. In other words, the $qv$ vector so defined is a feasible solution to the first program below, justifying one inequality:

$$\epsilon'' \geq \max_{\pi,\|\eta\|_{\Sigma_{pk}}\leq\sqrt{\sigma_e}} \left[\overline{\phi}_{\pi,t}^\top\eta\right] \geq \max_\pi \left[\overline{\phi}_{\pi,t}^\top(qv)\right] = \|qv\|_2 \max_\pi\left(\overline{\phi}_{\pi,t}^\top\frac{(qv)}{\|qv\|_2}\right) \tag{141}$$

$$\geq \|qv\|_2\nu_{min} = q\nu_{min} = \sqrt{\frac{\sigma_e}{\lambda_{min}(\Sigma_{pk})}}\nu_{min}. \tag{142}$$

Solving for $\lambda_{min}$ gives:

$$\lambda_{min}(\Sigma_{pk}) \geq \sigma_e \left(\frac{\nu_{min}}{\epsilon}\right)^2 \geq \sigma_e \times 2 \times 8d_p \ln \frac{2d_p}{\delta''} = \sigma_{e+1} \times 8d_p \ln \frac{2d_p}{\delta''} \tag{143}$$

Therefore the inductive hypothesis must hold for $e + 1$ as well, in other words, the statement in inductive hypothesis 4 must hold for all $e$.

Now we determine the required value for $\rho$ at the end of the phase. We want to ensure

$$\max_{\pi,\eta\in\mathbb{R}^{d_p}:\|\eta\|_{\Sigma_{pk}}\leq\sqrt{\alpha_p}} \overline{\phi}_{\pi,p}^\top \eta \leq \frac{\epsilon}{2H} \tag{144}$$

where now $k$ *is the episode at the end of phase* $p$. Since the inductive hypothesis holds in epoch $e$, lemma 16 ensures

$$\max_{\pi,\eta\in\mathbb{R}^{d_p}:\|\eta\|_{\Sigma_{pk}}\leq\sqrt{\sigma}} \overline{\phi}_{\pi,p}^\top \eta \leq \epsilon''. \tag{145}$$

We combine the above finding with a scaling argument given by lemma 17 *(Uncertainty Lemma)* that gives:

$$\max_{\pi,\eta\in\mathbb{R}^{d_p}:\|\eta\|_{\Sigma_{pk}}\leq\sqrt{\alpha_p}} \overline{\phi}_{\pi,p}^\top \eta = \sqrt{\frac{\alpha_p}{\sigma_e}} \times \left( \max_{\pi,\eta\in\mathbb{R}^{d_p}:\|\eta\|_{\Sigma_{pk}}\leq\sqrt{\sigma_e}} \overline{\phi}_{\pi,p}^\top \eta \right) \leq \sqrt{\frac{\alpha_p}{\sigma_e}} \epsilon''. \tag{146}$$

Requiring the above rhs to be $\leq \frac{\epsilon}{2H}$ gives a condition on the number of epochs $e_{max}$ required ($e_{max}$ is the number of epochs) and on $\sigma_{e_{max}}$; setting $\epsilon'' = \epsilon$ gives

$$\sqrt{\frac{\alpha_p}{\sigma_{e_{max}}}}\epsilon \leq \frac{\epsilon}{2H} \rightarrow \sqrt{\frac{\sigma_{e_{max}}}{\alpha_p}} \geq 2H \tag{147}$$

$$\rightarrow \sigma_{e_{max}} = 2^{e_{max}-1}\sigma_{Start} \geq 4H^2\alpha_p \quad \text{(by induction)} \tag{148}$$

$$\rightarrow 2^{e_{max}-1} \geq \frac{4H^2\alpha_p}{\sigma_{Start}} \rightarrow e_{max} = \left\lceil 1 + \ln_2\left(\frac{4H^2\alpha_p}{\sigma_{Start}}\right)\right\rceil. \tag{149}$$

In every epoch, $\epsilon'' = \epsilon$ and so the number of episodes necessary to achieve the required precision is (see lemma 16 *(Learning an Epoch)*):

$$\sum_{e=1}^{e_{max}} \left\lceil \frac{2}{1-q} \times \frac{(\sqrt{\gamma_t(\sigma_e)D_p} + A)^2}{\epsilon^2} \right\rceil \tag{150}$$

and since $\gamma_t(\sigma_e)$ strictly increases with $e$ we can say that

$$\left\lceil \frac{2}{1-q} \times \frac{(\sqrt{\gamma_t(\sigma_{e_{max}})D_p} + A)^2}{\epsilon^2} \right\rceil \times e_{max} \tag{151}$$

episodes suffices. $\qquad\square$

## D.5 Learning to Navigate

In this section we show that FRANCIS "learns to navigate", minimizing the least-square error in LSVI across timesteps.

**Proposition 3** (Learning to Navigate). *Assume that[11]:*

1. $\mathcal{I}(\mathcal{Q}_t, \mathcal{Q}_{t+1}) \leq \frac{\epsilon}{2H}$    *(this is always satisfied by our assumptions on $\epsilon$)*

2. $\left(\frac{\nu}{\epsilon}\right)^2 \geq 2 \times 8d_p \ln \frac{2d_p}{\delta''}$    *(this is also always satisfied by our assumptions on $\epsilon$)*

*Then after*

$$\widetilde{O}\left(H^2 \sum_{t=1}^{H} \frac{d_t^2(d_t + d_{t+1})}{\epsilon^2}\right) \tag{152}$$

*episodes, outside of the failure event it holds that*

$$\sum_{t=1}^{H}\left[\mathcal{I}(\mathcal{Q}_t, \mathcal{Q}_{t+1}) + \sqrt{\alpha_t}\|\overline{\phi}_{\pi,t}\|_{\Sigma_t^{-1}}\right] \leq \epsilon, \quad \forall \pi \tag{153}$$

*and in particular*

$$\mathcal{I}(\mathcal{Q}_t, \mathcal{Q}_{t+1}) + \sqrt{\alpha_t}\|\overline{\phi}_{\pi,t}\|_{\Sigma_t^{-1}} \leq \frac{\epsilon}{H}, \quad \forall \pi, t \in [H]. \tag{154}$$

*Proof.* We proceed by induction over timesteps / phases $p$:

**Inductive Hypothesis 5** (Main Inductive Hypothesis). *In phase $p \in [H]$ it holds that*

1. $\sum_{t=1}^{p-1}\left[\mathcal{I}(\mathcal{Q}_t, \mathcal{Q}_{t+1}) + \sqrt{\alpha_t}\|\overline{\phi}_{\pi,t}\|_{\Sigma_t^{-1}}\right] \leq \frac{p-1}{H}\epsilon$    *(this ensures accuracy in LSVI)*

2. $\lambda_{min}(\Sigma_t) \geq 4H^2\alpha_t \quad t \in [p-1]$    *(this ensures boundness of the iterates in LSVI)*

The inductive hypothesis vacuously holds for $p = 1$ (there is nothing to check). Now we show the inductive step. Assume the inductive hypohesis holds for a generic $p - 1$, we want to show it still holds for $p$. A direct application of lemma 18 *(Learning a Level)* gives ($\Sigma_p$ is the covariance matrix after learning has completed):

$$\sqrt{\alpha_p}\|\overline{\phi}_{\pi,p}\|_{\Sigma_p^{-1}} \overset{\text{lemma } 26}{=} \max_{\pi,\eta\in\mathbb{R}^{d_p}:\|\eta\|_{\Sigma_p}\leq\sqrt{\alpha_p}} \overline{\phi}_{\pi,p}^{\top}\eta \leq \frac{\epsilon}{2H} \tag{155}$$

Adding

$$\mathcal{I}(\mathcal{Q}_t, \mathcal{Q}_{t+1}) \leq \frac{\epsilon}{2H} \tag{156}$$

to both sides and adding the result to the equation in the inductive hypothesis proves the inductive step. The final number of episodes follows from summing the episodes needed in every phases according to lemma 18 *(Learning a Level)*. □

## D.6 Solution Reconstruction (*Main Result*)

In this section we present our main result in a more formal way than in the main text; throughout the appendix the symbols are generally reported in table 2.

First, let us define the reward classes.

**Definition 6** (Reward Classes). *Consider an MDP $\mathcal{M}(\mathcal{S}, \mathcal{A}, p, \cdot, H)$ without any reward function. Fix a misspecification function $\Delta_t^r(\cdot, \cdot, ) : \mathcal{S} \times \mathcal{A} \to \mathbb{R}$ for every $t \in [H]$ which can depend on the state and action pair, and is subject to the constraint*

$$\forall (\pi, t) \quad |\mathbb{E}_{x_t \sim \pi} \Delta_t^r(x_t, \pi_t(x_t))| \stackrel{def}{=} |\overline{\Delta}_{\pi, t}^r| \leq E_t. \tag{157}$$

*Define the following class $\mathfrak{I}$ (Implicit Regularity) of (expected) reward functions $(r_1, \ldots, r_H)$ on $\mathcal{M}$, parameterized by $(\theta_1^r, \ldots, \theta_H^r)$ and satisfying $\forall (s, a, t, \pi) \in \mathcal{S} \times \mathcal{A} \times [H] \times \Pi$ (here $\Pi$ is the policy space):*

1. $r_t(s, a) = \phi_t(s, a)^\top \theta_t^r + \Delta_t^r(s, a)$

2. $|\Delta_t^r(s, a)| \leq 1$

3. $|\mathbb{E}_{x_t \sim \pi} r_t(x_t, \pi_t(x_t))| \leq \frac{1}{H}$

*In addition, define the following class $\mathfrak{E}$ (Explicit Regularity) of (expected) reward functions $(r_1, \ldots, r_H)$ on $\mathcal{M}$ parameterized by $(\theta_1^r, \ldots, \theta_H^r)$ satisfying $\forall (s, a, t, \pi) \in \mathcal{S} \times \mathcal{A} \times [H] \times \Pi$:*

1. $r_t(s, a) = \phi_t(s, a)^\top \theta_t^r + \Delta_t^r(s, a)$

2. $|\Delta_t^r(s, a)| \leq 1$

3. $\|\theta_t^r\|_2 \leq \frac{1}{H}$.

Under explicit regularity the bound on $\|\theta_t^r\|_2$ constrains the maximum value the reward can take; instead, under implicit regularity we do not have such requirement, as only the expectation is controlled. This implies the local reward can be much larger than the expectation, making this a much harder setting.

We are now ready to present the main result formally.

**Theorem 1** (Restating theorem 4.1 formally). *Consider an MDP $\mathcal{M}$ and a feature extractor $\phi$ satisfying $\|\phi_t(s, a)\|_2 \leq 1$ for every $(s, a) \in \mathcal{S} \times \mathcal{A}$ and fix two classes of reward functions $\mathfrak{I}$ and $\mathfrak{E}$ according to definition 6 (Reward Classes). Set $\epsilon$ to satisfy $\epsilon \geq \widetilde{\Omega}(d_t H(\mathcal{I}(\mathcal{Q}_t, \mathcal{Q}_{t+1}) + E_t))$ and $\epsilon \leq \widetilde{O}(\nu_{min}/\sqrt{d_t})$ for all $t \in [H]$.*

*FRANCIS always terminates after $\widetilde{O}\left(H^2 \sum_{t=1}^H \frac{d_t^2(d_t + d_{t+1})}{\epsilon^2}\right)$ episodes (with probability one), returning a dataset $\mathcal{D} = \{(s_{tk}, a_{tk}, s_{t+1,k}^+)\}_{t=1, \ldots, H}^{k=1, \ldots, n(t)}$ of the collected state-action-successor states $(s_{tk}, a_{tk}, s_{t+1,k}^+)$ in episode $k \in [n(t)]$ for each timestep $t \in [H]$.*

*Now consider any reward function $r \in \mathfrak{E}$ or $r \in \mathfrak{I}$ and the MDP induced by that reward function $\mathcal{M}(\mathcal{S}, \mathcal{A}, p, r, H)$, and replace each tuple $(s_{tk}, a_{tk}, s_{t+1,k}^+) \in \mathcal{D}$ with $(s_{tk}, a_{tk}, r_{tk}, s_{t+1,k}^+)$ where $r_{tk}$ satisfies*

$$r_{tk} = r_t(s_{tk}, a_{tk}) + \eta^r \tag{158}$$

*where $\eta^r$ is 1-sub-Gaussian noise.*

*Then with probability at least $1 - \delta$, the batch LSVI algorithm run on $\mathcal{D}$ (see algorithm 3) returns a policy $\pi$ such that on $\mathcal{M}$*

$$\mathbb{E}_{x_1 \sim \rho}(V_1^\star - V_1^\pi)(x_1) \leq \frac{\epsilon}{\nu_{min}}. \tag{159}$$

*if $r \in \mathfrak{I}$ and*

$$\mathbb{E}_{x_1 \sim \rho}(V_1^\star - V_1^\pi)(x_1) \leq \epsilon. \tag{160}$$

*if $r \in \mathfrak{E}$.*

We have expressed the theorem in its full generality, but if the reward function is prescribed a posteriori through an oracle then we expect the noise $\eta^r$ in eq. (158) to be absent. In general, if the reward function is prescribed a posteriori then it should be prescribed as a linear function (in the chosen features) to avoid any additional error in the LSVI procedure. Finally the reward misspecification $\Delta_t^r(\cdot, \cdot)$ can depend on the parameter $\theta$ if it is a Lipshitz function of $\theta$. Alternatively, if it is a discontinuous function of $\theta$ then same-order guarantees are still recovered if eq. (157) is replaced with $\forall (s, a, t) \quad |\Delta_t^r(s, a)| \leq E_t$.

*Proof.* (of the main result) Let $n(t)$ the number of samples collected at each level (notice that we only store one sample every trajectory, so the number of samples equals the number of trajetories / number of episodes), according to lemma 18 *(Learning a Level)*. Using the assumptions on $\epsilon$ (these conditions are used in the good event for LSVI in definition 5 *(Good Event for* LSVI*)*) we can ensure:

$$\sqrt{n(t)}E_t = \sqrt{\frac{n(t)}{\alpha_t}}E_t\sqrt{\alpha_t} = \widetilde{O}\left(\frac{d_t H\sqrt{\alpha_t}}{\sqrt{\alpha_t}\epsilon}\right)E_t\sqrt{\alpha_t} \leq \sqrt{\alpha_t}/3 \quad (161)$$

$$\sqrt{n(t)}\mathcal{I}(\mathcal{Q}_t, \mathcal{Q}_{t+1}) = \sqrt{\frac{n(t)}{\alpha_t}}\mathcal{I}(\mathcal{Q}_t, \mathcal{Q}_{t+1})\sqrt{\alpha_t} = \widetilde{O}\left(\frac{d_t H\sqrt{\alpha_t}}{\sqrt{\alpha_t}\epsilon}\right)\mathcal{I}(\mathcal{Q}_t, \mathcal{Q}_{t+1})\sqrt{\alpha_t} \leq \sqrt{\alpha_t}/3. \quad (162)$$

We assume we are in the good event[12] for FRANCIS, see definition 7 *(Good Event for* FRANCIS*)*, which occurs with probability $1 - \delta$ according to lemma 19 *(Probability of Good Event for* FRANCIS*)*. We apply proposition 3 *(Learning to Navigate)*, which gives the stated number of episodes to termination and the condition satisfied by the samples in the dataset $\mathcal{D}$ (through the covariance matrices $\Sigma_t^{-1}$):

$$\sum_{t=1}^{H}\left[\mathcal{I}(\mathcal{Q}_t, \mathcal{Q}_{t+1}) + \sqrt{\alpha_t}\|\overline{\phi}_{\pi,t}\|_{\Sigma_t^{-1}}\right] \leq \epsilon, \quad \forall\pi$$

$$\mathcal{I}(\mathcal{Q}_t, \mathcal{Q}_{t+1}) + \sqrt{\alpha_t}\|\overline{\phi}_{\pi,t}\|_{\Sigma_t^{-1}} \leq \frac{\epsilon}{H}, \quad \forall\pi, t \in [H]. \quad (163)$$

Now, under *implicit regularity* lemma 20 *(Reward Boundness)* ensures (the lemma requires $E_t \leq \frac{1}{H}$, which is always satisfied since we must have $\epsilon < 1$ to produce any useful result, and from the theorem hypothesis $E_t \leq \epsilon/(d_t H) \leq 1/H$)

$$\|\theta_t^R\|_2 \leq \frac{2}{H\nu_{min}} \overset{def}{=} \frac{R}{H}. \quad (164)$$

Finally, proposition 2 *(Batch* LSVI *Guarantees (algorithm 3))* ensures that LSVI in algorithm 3 returns a value function $\widehat{V}$ and policy $\widehat{\pi}^\star$ such that

$$\mathbb{E}_{x_1 \sim \rho}\left(V_1^\star - \widehat{V}_1\right)(x_1) \leq \sum_{t=1}^{H}\left[2E_t + R\left(\mathcal{I}(\mathcal{Q}_t, \mathcal{Q}_{t+1}) + \sqrt{\alpha_t}\|\overline{\phi}_{\pi^\star,t}\|_{\Sigma_t^{-1}}\right)\right]$$

$$\mathbb{E}_{x_1 \sim \rho}\left(\widehat{V}_1 - V_1^{\widehat{\pi}^\star}\right)(x_1) \leq \sum_{t=1}^{H}\left[2E_t + R\left(\mathcal{I}(\mathcal{Q}_t, \mathcal{Q}_{t+1}) + \sqrt{\alpha_t}\|\overline{\phi}_{\widehat{\pi}^\star,t}\|_{\Sigma_t^{-1}}\right)\right]. \quad (165)$$

Using eq. (163) (and recalling $E_t \leq \epsilon$ by hypothesis of the theorem) to further simplify it we obtain:

$$\mathbb{E}_{x_1 \sim \rho}\left(V_1^\star - \widehat{V}_1\right)(x_1) \leq 2R\epsilon$$

$$\mathbb{E}_{x_1 \sim \rho}\left(\widehat{V}_1 - V_1^{\widehat{\pi}^\star}\right)(x_1) \leq 2R\epsilon.$$

Summing the two expression gives:

$$\mathbb{E}_{x_1 \sim \rho}\left(V_1^\star - V_1^{\widehat{\pi}^\star}\right)(x_1) \leq 4R\epsilon.$$

Rescaling $\epsilon$ by 4 and substituting the value for $R$ gives the thesis under *implicit regularity*.

Under *explicit regularity* the steps are the same, but now

$$\|\theta_t^r\|_2 \leq \frac{1}{H} \overset{def}{=} \frac{R}{H} \quad (166)$$

is explicitly prescribed, and the thesis immediately follows. □

The generality of the main result allows us to immediately obtain the following corollary:

**Corollary 1** (Learning a Prescribed Reward Function during the Execution). *Under the same assumptions as theorem 1, assume the reward function $r \in \mathfrak{E}$ or $r \in \mathfrak{I}$ is prescribed before the execution of* FRANCIS *and*

$$r_{tk} = r_t(s_{tk}, a_{tk}) + \eta^r \quad (167)$$

*where $\eta^r$ is 1-sub-Gaussian noise. Assume $(s_{tk}, a_{tk}, r_{tk}, s_{t+1,k}^+)$ is stored in the dataset $\mathcal{D}$.*

*Then with probability at least $1 - \delta$, the batch* LSVI *algorithm run on $D$ (see algorithm 3) returns a policy $\pi$ such that on $\mathcal{M}$*

$$\mathbb{E}_{x_1 \sim \rho}(V_1^\star - V_1^\pi)(x_1) \leq \frac{\epsilon}{\nu_{min}}. \tag{168}$$

*if $r \in \mathfrak{I}$ and*

$$\mathbb{E}_{x_1 \sim \rho}(V_1^\star - V_1^\pi)(x_1) \leq \epsilon. \tag{169}$$

*if $r \in \mathfrak{E}$.*

## D.7 Computational Complexity

Theorem 4.1 gives a bound on the number of episodes to termination. In every episode, a multivariate normal vector is sampled (which can be done efficiently) and LSVI is invoked.

Assume $d_1 = \cdots = d_H = d$ for simplicity; a naive implementation would factorize and store the new covariance matrix at the end of a phase (total of $\widetilde{O}(Hd^3)$ work across all phases); after this, computing the $\widehat{\theta}_t$'s requires $\widetilde{O}\left(H(d^2 + Ad) \times n_{episodes}\right)$ computations at every episode where $n_{episodes}$ is the total number of episodes at termination given in theorem 4.1.

**Definition 7** (Good Event for FRANCIS). *We say the good event for* FRANCIS *occurs if for all timesteps $t \in [H]$ or phases $p \in [H]$ and episodes $k$ in that phase the following bounds[13] jointly hold and we are in the good event for* LSVI *(see definition 5 (Good Event for* LSVI*)).*

$$\left| \frac{1}{i_{max}} \sum_{i=1}^{i_{max}} \zeta_{pk(e,i)} \right| \leq \sqrt{\frac{2(2L_\phi \mathcal{R}_t)^2 \ln\left(\frac{1}{\delta''}\right)}{i_{max}}} = \frac{\sqrt{8 \ln\left(\frac{1}{\delta''}\right)}}{\sqrt{i_{max}}} \overset{def}{=} \frac{A}{\sqrt{i_{max}}} \tag{170}$$

$$\|\xi_{t,k(e,i)}\|_{\Sigma_{t,k(e,i)}} \leq \sqrt{\gamma_t(\sigma)} \overset{def}{=} \sqrt{2\sigma_t d_t \ln \frac{2d_t}{\delta''}} \tag{171}$$

$$\|\xi_{t,k(e,i)}\|_2 \leq \sqrt{\frac{2\sigma_t d_t}{\lambda_{min}(\Sigma_{p,k(e,i)})} \ln \frac{2d_t}{\delta''}} \tag{172}$$

$$\frac{1}{k_{max}} \sum_{k=1}^{k_{max}} \mathbb{1}\{\mathcal{E}_k \mid \mathcal{C}_k\} \geq (1-q) - \sqrt{\frac{2\ln\left(\frac{1}{\delta''}\right)}{k_{max}}} \tag{173}$$

**Lemma 19** (Probability of Good Event for FRANCIS). *There exists a parameter $\delta'' = \frac{\delta}{poly(d_1,\dots,d_H,H,\frac{1}{\epsilon})}$, such that the good event of definition 7 holds with probability at least $1 - \delta$.*

*Proof.* The first and fourth inequality follow from lemma 24 *(Azuma-Hoeffding Inequality)*. The second and third inequality follow from lemma 22 *(Large Deviation Multivariate Normal)*. In particular, a union bound over the statements, over $H$ and over the number of episodes ensures all statements jointly hold at any point during the execution of the program; from this, the value for $\delta''$ can be determined. □

# E  Lower Bound

We sketch the lower bound to highlight that explorability is required.

**Proposition 4** (Lower Bound on Explorability Dependence under Implicit Regularity). *There exists an MDP and a feature map $\phi_t : (s, a) \mapsto \phi_t(s, a) \in \mathbb{R}^2$ with explorability parameter $\nu_{min}$ and a reward function such that:*

$$\forall(\pi, t) \quad r_t(s, a) = \phi_t(s, a)^\top \theta_t^r, \quad |\mathbb{E}_{x_t \sim \pi} r_t(x_t, \pi_t(x_t))| \leq 1 \tag{174}$$

*and yet no reinforcement learning agent without knowledge of $\theta^r$ can return an $\epsilon$-optimal policy for $\epsilon \leq \nu_{min} \leq \frac{1}{4}$ in less than $\Omega(1/(\epsilon \nu_{min})^2)$ trajectories with probability higher than $2/3$.*

Notice that the proposition above is for a fixed (but unknown) deterministic reward function; this is thus a special case of the reward-free learning setting we consider, implying that the hardness is due to the implicit regularity conditions rather than to reward-free learning.

The proof essentially uses a multi-armed bandit lower bound where the noise is $1/\nu_{min}$-sub-Gaussian and is created using the MDP dynamics (since the reward is deterministic).

*Proof.* We construct the MDP as follows: there is a single starting state $s_{start}$ with two actions $a_L$ and $a_R$ and the identity feature $\phi_1(s_{start}, a_L) = e_1, \phi_1(s_{start}, a_R) = e_2$, where $e_1, e_2$ are canonical vectors in $\mathbb{R}^2$. Now fix a scalar $\epsilon \in [-\frac{\nu_{min}}{2}, \frac{\nu_{min}}{2}]$:

1. action $a_L$ gives an immediate reward $-1/2$ and leads to state $s_{L1}$ with probability $\frac{1}{2} + \nu_{min}$ and to $s_{L2}$ with probability $\frac{1}{2} - \nu_{min}$. The feature map reads $\phi_2(s_{L1}) = e_1$ and $\phi_2(s_{L2}) = -e_1$ in the only action available in each state.

2. action $a_R$ gives an immediate reward $-1/2$ and leads to state $s_{R1}$ with probability $\frac{1}{2} + \nu_{min} + \epsilon$ and to $s_{R2}$ with probability $\frac{1}{2} - \nu_{min} - \epsilon$. The feature map reads $\phi_2(s_{R1}) = e_2$ and $\phi_2(s_{R1}) = -e_2$

In this MDP there are only two distinct policies: $\pi_L$ that selects $a_L$ first and then the only available action in either $s_{L1}$ or $s_{L2}$, and $\pi_R$ that selects $a_R$ first and then the only available action in either $s_{R1}$ or $s_{R2}$. Therefore, this is equivalent to a *multiarmed* bandit problem with reward $-1/2 + \overline{\phi}_{\pi_L,2}^\top \theta_2^r$ for $\pi_L$ and $-1/2 + \overline{\phi}_{\pi_R,2}^\top \theta_2^r$ for $\pi_2$. The minimum explorability coefficient is ($\nu_1 = 1$ at timestep 1)

$$\min_{\theta \neq 0} \max_\pi \overline{\phi}_{\pi,2}^\top \frac{\theta}{\|\theta\|_2} = \left[\left(\frac{1}{2} + \nu_{min} - \frac{\nu_{min}}{2}\right) - \left(\frac{1}{2} - \nu_{min} + \frac{\nu_{min}}{2}\right)\right] e_2^\top e_2 = \nu_{min} \tag{175}$$

corresponding to policy $\pi_R$ (this can be computed by inspection; notice that $\pi_L$ yields the same $\nu_{min}$). Now consider the reward parameter $\theta_2^r = 1/\nu_{min} \times [1/2, 1/2]$; the expected reward at timestep 2 under policy $\pi_R$ is $\mathbb{E}_{x_2 \sim \pi_L} r_2(x_2) = \nu_{min} \times \frac{1}{2\nu_{min}} \leq 1$ which satisfies the assumptions of the lemma. At the same time $\mathbb{E}_{x_2 \sim \pi_R} r_2(x_2) = (\nu_{min} + 2\epsilon) \times \frac{1}{2\nu_{min}} \leq 1$. This implies the random return $-1/2 + \phi_2(s)^\top \theta_2$ with $s \sim p_1(s_{start}, a_L)$ is a scaled and shifted Bernoulli random variable with mean zero, taking the values $-1/2 + \frac{1}{2\nu_{min}}$ and $-1/2 - \frac{1}{2\nu_{min}}$. Since the standard deviation of this random variables (with $\nu_{min} \leq \frac{1}{4}$) is $\Omega(1/\nu_{min})$, this random variable must be $\Omega(1/\nu_{min})$-sub-Gaussian[14]. The same reasoning applies to $-1/2 + \phi_2(s)^\top \theta_2$ with $s \sim p_1(s_{start}, a_R)$. Notice that both expectations are at most 1.

Solving this class of problems (parameterized by $\epsilon$), i.e., identifying an $|\epsilon|/2$-optimal policy is equivalent to solving a multiarmed bandit problem with 2 actions (corresponding to the policies $\pi_1$ and $\pi_2$). This construction is exactly the same as theorem 2 from Krishnamurthy et al. [2016] with shifted Bernoulli random variables that are scaled by the inverse explorability coefficient $1/\nu_{min}$. This implies that a sample complexity $\Omega(1/(\nu_{min}|\epsilon|)^2)$ is required to output an $|\epsilon|/2$-optimal policy with probability $> 2/3$. $\square$

# F   Support Lemmas

**Lemma 20** (Reward Boundness). *If we assume that*

$$\forall \pi \qquad |\mathbb{E}_{x_t \sim \pi} r_t(x_t, \pi_t(x_t))| \leq \frac{1}{H} \tag{176}$$

$$and \quad \exists \theta_t^r \in \mathbb{R}^{d_t} \quad such\ that \qquad |\mathbb{E}_{x_t \sim \pi} r_t(x_t, \pi_t(x_t)) - \overline{\phi}_{\pi,t}^\top \theta_t^r| \leq E_t \leq \frac{1}{H} \tag{177}$$

*then it follows that*

$$\|\theta_t^r\|_2 \leq \frac{2}{H\nu_t}. \tag{178}$$

*Proof.* From the hypothesis it follows

$$\frac{2}{H} \geq |\overline{\phi}_{\pi,t}^\top \theta_t^r| = \|\theta_t^r\|_2 \times |\overline{\phi}_{\pi,t}^\top \frac{\theta_t^r}{\|\theta_t^r\|_2}| \tag{179}$$

in particular this must hold for the policy $\pi$ that maximizes the above display. Therefore, after taking $\max_\pi$, take $\min_{\|\theta\|_2=1}$ to obtain (using definition 2 *(Explorability)*):

$$\geq \|\theta_t^r\|_2 \times \min_{\|\theta\|_2=1} \max_\pi |\overline{\phi}_{\pi,t}^\top \theta| = \|\theta_t^r\|_2 \nu_t. \tag{180}$$

Rearranging

$$\|\theta_t^r\|_2 \leq \frac{2}{H\nu_t}. \tag{181}$$

$\square$

## F.1 High Probability Bounds

**Lemma 21** (Transition Noise High Probability Bound). *If $\lambda = 1$ and $R = 2L_\phi \mathcal{R}_{t+1}$ with probability at least $1 - \delta'$ it holds that $\forall V_{t+1} \in \mathcal{V}_{t+1}$:*

$$\left\| \sum_{i=1}^{k-1} \phi_{ti} \left( V_{t+1}(s_{t+1,k}^+) - \mathbb{E}_{s' \sim p(s_{tk}, a_{tk})} V_{t+1}(s') \right) \right\|_{\Sigma_t^{-1}} \leq \sqrt{\beta_t^t} \tag{182}$$

*where:*

$$\sqrt{\beta_t^t} \overset{def}{=} \sqrt{2} \times 2 \sqrt{\frac{d_t}{2} \ln\left(1 + L_\phi^2 k/d_t\right) + d_{t+1} \ln(1 + 4\mathcal{R}_{t+1}/(2L_\phi\sqrt{k})) + \ln\left(\frac{1}{\delta'}\right)} + 2. \tag{183}$$

*Proof.* Since the statement needs to hold for every $V_{t+1} \in \mathcal{V}_{t+1}$, we start by constructing an $\epsilon$-cover for set $\mathcal{V}_{t+1}$ using the supremum distance. To achieve this, we construct an $\epsilon$-cover for the parameter $\theta \in \mathcal{B}_{t+1}$ using the "Covering Number of Euclidean Ball" lemma in [Zanette et al., 2020b]. This ensures that there exists a set $\mathcal{D}_{t+1} \subseteq \mathcal{B}_{t+1}$, containing $(1 + 2\mathcal{R}_{t+1}/\epsilon')^{d_{t+1}}$ vectors $\overset{\triangle}{\theta}_{t+1}$ that well approximates any $\theta_{t+1} \in \mathcal{B}_{t+1}$:

$$\exists \mathcal{D}_{t+1} \subseteq \mathcal{B}_{t+1} \quad \text{such that} \quad \forall \theta_{t+1} \in \mathcal{B}_{t+1}, \quad \exists \overset{\triangle}{\theta}_{t+1} \in \mathcal{D}_{t+1} \quad \text{such that} \quad \|\theta_{t+1} - \overset{\triangle}{\theta}_{t+1}\|_2 \leq \epsilon'. \tag{184}$$

Let $\overset{\triangle}{V}_{t+1}(s) \overset{def}{=} \max_a \phi_{t+1}(s,a)^\top \overset{\triangle}{\theta}$, where $\overset{\triangle}{\theta} = \arg\min_{\theta' \in \mathcal{D}_{t+1}} \|\theta' - \theta\|_2$ and consider $V_{t+1} \in \mathcal{V}_{t+1}$. For any fixed $s \in \mathcal{S}$ we have that:

$$\begin{aligned}
\left| \left(V_{t+1} - \overset{\triangle}{V}_{t+1}\right)(s) \right| &= \left| \max_{a'} \phi_{t+1}(s, a')^\top \theta_{t+1} - \max_{a''} \phi_{t+1}(s, a'')^\top \overset{\triangle}{\theta}_{t+1} \right| \\
&\leq \max_a \left| \phi_{t+1}(s, a)^\top \left( \theta_{t+1} - \overset{\triangle}{\theta}_{t+1} \right) \right| \\
&\leq \max_a \|\phi_{t+1}(s, a)\|_2 \|\theta_{t+1} - \overset{\triangle}{\theta}_{t+1}\|_2 \\
&\leq L_\phi \epsilon'.
\end{aligned} \tag{185}$$

By using the triangle inequality we can write:

$$\begin{aligned}
&\left\| \sum_{i=1}^{k-1} \phi_{ti} \left( V_{t+1}(s_{t+1,k}^+) - \mathbb{E}_{s' \sim p(s_{tk}, a_{tk})} V_{t+1}(s') \right) \right\|_{\Sigma_t^{-1}} \\
&\leq \left\| \sum_{i=1}^{k-1} \phi_{ti} \left( \overset{\triangle}{V}_{t+1}(s_{t+1,k}^+) - \mathbb{E}_{s' \sim p(s_{tk}, a_{tk})} \overset{\triangle}{V}_{t+1}(s') \right) \right\|_{\Sigma_t^{-1}} + \\
&\quad + \left\| \sum_{i=1}^{k-1} \phi_{ti} \left( \mathbb{E}_{s' \sim p(s_{tk}, a_{tk})} \overset{\triangle}{V}(s') - \mathbb{E}_{s' \sim p(s_{tk}, a_{tk})} V_{t+1}(s') \right) \right\|_{\Sigma_t^{-1}} \\
&\quad + \left\| \sum_{i=1}^{k-1} \phi_{ti} \left( V_{t+1}(s_{t+1,k}^+) - \overset{\triangle}{V}_{t+1}(s_{t+1,k}^+) \right) \right\|_{\Sigma_t^{-1}}.
\end{aligned} \tag{186}$$

Each of the last two terms above can be written for some $b_i$'s (different for each of the two terms) as $\left\| \sum_{i=1}^{k-1} \phi_{ti} b_i \right\|_{\Sigma_{tk}^{-1}}$. The projection lemma, (lemma 8 from Zanette et al. [2020b]) ensures:

$$\left\| \sum_{i=1}^{k-1} \phi_{ti} b_i \right\|_{\Sigma_t^{-1}} \leq L_\phi \epsilon' \sqrt{k} \tag{187}$$

We have used eq. (185) to bound the $b_i$'s. Now we examine the first term of the rhs in equation in eq. (186). In particular, we bound that term for a generic $\overset{\triangle}{V}_{t+1}$ and then do a union bound over all possible $\overset{\triangle}{V}_{t+1}$, which are generated by finitely many $\overset{\triangle}{\theta}_{t+1} \in \mathcal{D}_{t+1}$ as explained before. We obtain that:

$$\mathbf{P}\left( \bigcup_{\overset{\triangle}{\theta}_{t+1} \in \mathcal{D}_{t+1}} C(\overset{\triangle}{\overline{\theta}}_{t+1}) \right) \leq \sum_{\overset{\triangle}{\theta}_{t+1} \in \mathcal{D}_{t+1}} \mathbf{P}\left( C(\overset{\triangle}{\overline{\theta}}_{t+1}) \right) \leq (1 + 2\mathcal{R}_{t+1}/\epsilon')^{d_{t+1}} \delta'' \overset{def}{=} \delta' \tag{188}$$

where $C$ is the event reported below (along with $\delta''$) and the last inequality above follows from Theorem 1 in [Abbasi-Yadkori et al., 2011] (the random variables $\overset{\triangle}{V}_{t+1}(\cdot)$ and $\widehat{V}_{t+1}(\cdot)$ are $R = 2L_\phi \mathcal{R}_{t+1}$-subgaussian by construction):

$$C(\overset{\triangle}{\overline{\theta}}_{t+1}) \overset{def}{=} \left\{ \left\| \sum_{i=1}^{k-1} \phi_{ti} \left( \overset{\triangle}{V}_{t+1,i} - \mathbb{E}_{s' \sim p(s_{tk},a_{tk})} \overset{\triangle}{V}(s') \right) \right\|^2_{\Sigma_t^{-1}} > 2 \times (R)^2 \ln \left( \frac{\det(\Sigma_t)^{\frac{1}{2}} \det(\lambda I)^{-\frac{1}{2}}}{\delta''} \right) \right\}. \tag{189}$$

In particular, we set

$$\delta'' = \frac{\delta'}{(1 + 2\mathcal{R}_{t+1}/\epsilon')^{d_{t+1}}} \tag{190}$$

from the prior display and so with probability $1 - \delta'$ (after a union bound over all possible $\overset{\triangle}{\theta}_{t+1} \in \mathcal{D}_{t+1}$) we have upper bounded eq. (186) by:

$$R\sqrt{2\ln \left( \frac{\det(\Sigma_t)^{\frac{1}{2}} \det(\lambda I)^{-\frac{1}{2}} (1 + 2\mathcal{R}_{t+1}/\epsilon')^{d_{t+1}}}{\delta'} \right)} + 2L_\phi \epsilon' \sqrt{k}. \tag{191}$$

If we now pick

$$\epsilon' = \frac{R}{2L_\phi \sqrt{k}} \tag{192}$$

we get:

$$R\sqrt{2\ln \left( \frac{\det(\Sigma_t)^{\frac{1}{2}} \lambda^{-\frac{d_t}{2}} (1 + 2\mathcal{R}_{t+1}/\epsilon')^{d_{t+1}}}{\delta'} \right)} + R \tag{193}$$

$$= \sqrt{2}R\sqrt{\frac{1}{2}\ln(\det(\Sigma_t)) - \frac{d_t}{2}\ln(\lambda) + d_{t+1}\ln(1 + 2\mathcal{R}_{t+1}/\epsilon') + \ln \left( \frac{1}{\delta'} \right)} + R \tag{194}$$

Finally, using the Determinant-Trace Inequality (see lemma 10 of [Abbasi-Yadkori et al., 2011]) we obtain $\det(\Sigma_{tk}) \le \left( \lambda + L_\phi^2 k/d_t \right)^{d_t}$ and so (with $\lambda = 1$)

$$\le \sqrt{2} \times 2\sqrt{\frac{d_t}{2}\ln \left( 1 + L_\phi^2 k/d_t \right) + d_{t+1}\ln(1 + 4\mathcal{R}_{t+1}/(2L_\phi\sqrt{k})) + \ln \left( \frac{1}{\delta'} \right)} + 2 \overset{def}{=} \sqrt{\beta_t^t}. \tag{195}$$

$\square$

## F.2 Known Results

**Lemma 22** (Large Deviation Multivariate Normal). *Let $\Sigma \in \mathbb{R}^{d \times d}$ be an spd matrix with minimum eigenvalue $\lambda > 0$ and let*

$$\xi \sim \mathcal{N}\left(0, \sigma \Sigma^{-1}\right) \tag{196}$$

*for a positive scalar $\sigma$. For any fixed $\phi \in \mathbb{R}^d$ with probability at least $1 - \delta'$:*

$$|\phi^\top \xi|^2 \leq \frac{\sigma \|\phi\|_2^2}{\lambda} \left(2d \ln \frac{2d}{\delta'}\right) \tag{197}$$

*and so by choosing $\phi = \frac{\xi}{\|\xi\|_2}$ when $\xi \neq 0$ it holds that*

$$\|\xi\|_2 \leq \sqrt{\frac{\sigma}{\lambda} \left(2d \ln \frac{2d}{\delta'}\right)}. \tag{198}$$

*Under the same event it holds that*

$$\|\xi\|_\Sigma \leq \sqrt{\sigma \left(2d \ln \frac{2d}{\delta'}\right)}. \tag{199}$$

*Proof.* If

$$\xi \sim \mathcal{N}\left(0, \sigma \Sigma^{-1}\right) \tag{200}$$

it follows that

$$\frac{1}{\sqrt{\sigma}} \Sigma^{\frac{1}{2}} \xi \sim \mathcal{N}(0, I) \tag{201}$$

where $I$ is the identity matrix on $\mathbb{R}^d$. Therefore

$$\frac{1}{\sigma} \|\xi\|_\Sigma^2 = \left(\frac{1}{\sqrt{\sigma}} \xi^\top \Sigma^{\frac{1}{2}}\right)^\top \left(\frac{1}{\sqrt{\sigma}} \Sigma^{\frac{1}{2}} \xi\right) \sim \chi_d^2 \tag{202}$$

where $\chi_d^2$ is the chi-square distribution with $d$ degrees of freedom. From lemma 23 *($\chi$-square lemma)* we can compute a high probability bound for the above random variable (this also proves the last statement):

$$|\phi^\top \xi|^2 \leq \|\phi\|_{\Sigma^{-1}}^2 \|\xi\|_\Sigma^2 \leq \|\phi\|_2^2 \frac{\sigma}{\lambda} \frac{1}{\sigma} \|\xi\|_\Sigma^2 \leq \frac{\sigma \|\phi\|_2^2}{\lambda} \left(2d \ln \frac{2d}{\delta'}\right) \tag{203}$$

with probability at least $1 - \delta'$. $\square$

**Lemma 23** ($\chi$-square lemma). *Let $X^2 \sim \chi_d^2$ be a random variable that follows the chi-square distribution with $d$ degrees of freedom. With probability at least $1 - \delta'$*

$$X^2 \leq 2d \ln \frac{2d}{\delta'}. \tag{204}$$

*Proof.* Let $X_i \sim \mathcal{N}(0,1), i \in [d]$. If $X_i \in [-a, +a], \forall i \in [d]$ then it must follow that $\sum_{i \in [d]} X_i^2 \leq da^2$. Thus:

$$\mathbf{P}(X^2 = \sum_{i \in [d]} X_i^2 \geq da^2) \leq \mathbf{P}(\exists i \in [d], X_i \notin [-a, a]) = \mathbf{P}(\cup_{i \in [d]} X_i \notin [-a, a]) \leq d\,\mathbf{P}(X_i \notin [-a, a]) \leq 2de^{-a^2/2}.$$

Requiring the rhs above to be $\leq \delta'$ gives

$$a^2 = 2 \ln \frac{2d}{\delta'}.$$

$\square$

**Lemma 24** (Azuma-Hoeffding Inequality). *Let $X_i$ be a martingale difference sequence such that $X_i \in [-A, A]$ for some $A > 0$. Then with probability at least $1 - \delta'$ it holds that:*

$$\left| \sum_{i=1}^n X_i \right| \leq \sqrt{2A^2 n \ln \left(\frac{1}{\delta'}\right)}. \tag{205}$$

*Proof.* The Azuma inequality reads:

$$\mathbf{P}\left(\Big|\sum_{i=1}^{n} X_i\Big| \geq t\right) \leq e^{-\frac{2t^2}{4A^2 n}}, \tag{206}$$

see for example [Wainwright, 2019]. From here setting the rhs equal to $\delta'$ gives:

$$t \overset{def}{=} \sqrt{2A^2 n \ln\left(\frac{1}{\delta'}\right)}. \tag{207}$$

$\square$

**Lemma 25** (Change of $\Sigma$-Norm). *For a compatible vector $x \in \mathbb{R}^d$ and an spd matrix $\Sigma \in \mathbb{R}^{d \times d}$ with minimum eigenvalue $\lambda_{min}(\Sigma)$ we have*

$$\|x\|_{\Sigma} \geq \sqrt{\lambda_{min}(\Sigma)}\|x\|_2 \tag{208}$$

$$\|x\|_{\Sigma^{-1}} \leq \frac{1}{\sqrt{\lambda_{min}(\Sigma)}}\|x\|_2. \tag{209}$$

*Proof.* We show one inequality (the other is identical). Consider the eigendecomposition of $\Sigma$ with orthonormal eigenvectors $v_i$'s and eigenvalues $\lambda_i$'s:

$$\Sigma^{-1} = \sum_{i=1}^{d} \lambda_i^{-1} v_i v_i^{\top} \tag{210}$$

We can write:

$$\|x\|_{\Sigma^{-1}}^2 = x^{\top} \Sigma^{-1} x \tag{211}$$

$$= x^{\top} \left(\sum_{i=1}^{d} \lambda_i^{-1} v_i v_i^{\top}\right) x \tag{212}$$

$$= \sum_{i=1}^{d} \frac{1}{\lambda_i} \left(v_i^{\top} x\right)^2 \tag{213}$$

$$\leq \frac{1}{\lambda_{min}(\Sigma)} \sum_{i=1}^{d} \left(v_i^{\top} x\right)^2 \tag{214}$$

$$= \frac{1}{\lambda_{min}(\Sigma)} \|x\|_2^2. \tag{215}$$

$\square$

**Lemma 26** (Linear Bandit Exploration Bonus). *For an spd matrix $\Sigma$, the equality below holds whenever the operations make sense:*

$$\max_{\phi, \|\eta\|_{\Sigma} \leq \sqrt{\sigma}} \phi^{\top} \eta = \sqrt{\sigma}\|\phi\|_{\Sigma^{-1}} \tag{216}$$

*Proof.* Choose $\eta = \Sigma^{-1} \phi \frac{\sqrt{\sigma}}{\|\phi\|_{\Sigma^{-1}}}$, which satisfies the constraint

$$\|\Sigma^{-1}\phi \frac{\sqrt{\sigma}}{\|\phi\|_{\Sigma^{-1}}}\|_{\Sigma} = \|\phi \frac{\sqrt{\sigma}}{\|\phi\|_{\Sigma^{-1}}}\|_{\Sigma^{-1}} \sqrt{\sigma} = \sqrt{\sigma} \tag{217}$$

and gives an objective value

$$\max_{\phi, \|\eta\|_{\Sigma} \leq \sqrt{\sigma}} \phi^{\top} \eta \geq \phi \Sigma^{-1} \phi \frac{\sqrt{\sigma}}{\|\phi\|_{\Sigma^{-1}}} = \sqrt{\sigma}\|\phi\|_{\Sigma^{-1}} \tag{218}$$

On the other hand, Cauchy-Schwartz ensures:

$$\max_{\phi, \|\eta\|_{\Sigma} \leq \sqrt{\sigma}} \phi^{\top} \eta \leq \|\phi\|_{\Sigma^{-1}}\|\eta\|_{\Sigma} = \sqrt{\sigma}\|\phi\|_{\Sigma^{-1}}. \tag{219}$$

$\square$