[Reviews · NeurIPS 2020]

Review 1

Summary and Contributions: This paper propose a new RL algorithm that addressed the setting of low inherent Bellman error, which significantly improves the computational complexity of the earlier work in this setting [Zenette et al. 2020b]. This setting is also more general than the earlier line of work on linear MDP or low-rank MDP. This paper also present a reward-free version of the result where reward is not required in the phase of exploration

Strengths: The problem is important in the setting of linear function approximation. This result is solid, the contribution is significant, and the paper is well-written and easy to follow.

Weaknesses: This paper assumes the explorability (Definition 2). It seems to the lower bound requires the learning agent with no knowledge of reward vector, which is only justified in the reward-free setting? Can author comment on if this is necessary for the non-reward-free setting, and what is the analog of such condition in earlier result of linear MDP, such as LSVI-UCB, and if they require that?

Correctness: I have not fully checked the details. The high-level argument looks correct to me.

Clarity: Yes

Relation to Prior Work: Yes

Reproducibility: Yes

Additional Feedback:


Review 2

Summary and Contributions: The paper introduces a new provably efficient RL algorithm with PAC guarantees for linear MDPs in the low inherent Bellman error regime. The paper achieves this performance guarantee by a very simple variant of LSVI algorithm which basically relies on some approximation of G-optimal design to uniformly reduce the maximum uncertainty over all the features. This is simply done by keeping track of the online covariance matrix of the features and using that to generate reward from the corresponding Gaussian distribution. I believe if this result is proven to be true it is a novel and significant contribution to the literature of RL with function approximation and can be definitely of the great interest for NeurIPS community.

Strengths: The main strength of paper is that it tackles one of the major open problem in RL namely online RL with function approximation by producing a simple and computationally tractable algorithm that enjoys PAC guarantees in the agnostic reward setting. Beyond that the paper also considers the effect of approximation error, quantified by inherent Bellman error on the quality of results. I think the sheer fact that the paper has attempted on addressing such a difficult problem, which has eluded many top researchers in the past is admirable. Also although some of the assumptions in particular the explorability assumption looks very strong, the paper provides complementary results which shows these assumption in general are unavoidable. Overall I think a lot of work has gone into this paper and I believe this work definitely contributes to the literature of RL theory, even if not every result in the paper can be verified.

Weaknesses: The main difficulty is evaluating and verifying the results of the paper. In my belief The paper suffers from lack of proper and in-depth motivation for its theoretical and algorithmic arguments. This is the case both in description of Algorithm in Sect. 3 in which the paper fails to motivate the reasoning behind the proposed algorithmic steps. More importantly it is also the case in the sketch proof of section 5 which could significantly benefit from providing some high-level structure of proofs. In fact I found the current sketch proof very brief and unintuitive, which makes the job of evaluating the paper very difficult. Although by browsing through the sketch proof I couldn't find any error I was not able to reach to any overall understanding of the analysis of this paper based on the current version of sketch proof.

Correctness: I checked the sketch proof and the lower-bound on the exploitability and I couldn't find any error, I did not have time go through the full proof in the Appendix.

Clarity: As I mentions the lack of clarity is the main drawback of this work. I would suggest that authors t provide better motivations for their proposed approach and the analysis by describing the high level ideas in a more clear way.

Relation to Prior Work: The prior work in the literature of RL theory have been properly and extensively covered.

Reproducibility: No

Additional Feedback:


Review 3

Summary and Contributions: This paper provides a reward-agnostic pure exploration RL algorithm for systems with low-inherent Bellman error. The paper provides strong PAC guarantees on learning a near-optimal value function provided that the linear space is sufficiently “explorable”. The paper also shows a lower bound showing that the "explorable" condition is somewhat necessary. === After rebuttal: I do think the paper has merits in providing a reward free algorithm for the low-bellman error setting. But I still feel the reachability assumption is a bit too strong.

Strengths: (1) Low-inherent Bellman error is stronger than linear MDP or low-rank MDP. (2) Matching upper and lower bound. (3) Computational efficient algorithm.

Weaknesses: (1) The requirement for the "explorable" does not seem convincing for the following reasons: a. Theorem 4.1 requires that eps < O(\nu_min/sqrt{d}) b. The lower bound requires very small eps as well c. The lower bound appears to be built on a finite states MDP, in which case [Jin et al] provides an upper bound that has no dependence on \nu_min. Any explanation on this? d. Intuitively the \nu_min is not necessary in the sense that if a direction is hard to reach, then it is not important for any reward function. e. The algorithm is based on a layer-by-layer learning structure, which seems intrinsically requires a "explorable" condition, which is also observed in [Du et al. 2019]. f. When reduce to the tabular setting, it seems that this result does not recover [Jin et al] due to the requirement of \nu_min. (2) The layer-by-layer learning structure does not seem very natural. Such an algorithm may be loose on the H dependence. (3) There is a lack of motivation of discussing the low-Bellman error model and reward-free/agnositc exploration in this setting. (4) There is lack of experiment, although it seems fine for a theory-oriented paper.

Correctness: I believe so.

Clarity: It is okay written.

Relation to Prior Work: Yes.

Reproducibility: Yes

Additional Feedback:


Review 4

Summary and Contributions: The authors present new theoretical results for RL with linear function approximation, a setting for which there is still a lack of good results despite its seeming simplicity. The paper builds on results for batch learning RL algorithms for the low-inherent Bellman case as well as on recent work in G-optimal design. The main result is for reward-agnostic exploration where a linear reward function is revealed after data has been collected and a value function will be estimated in batch mode from the gathered experience. A conceptually simple algorithm is introduced for which they prove PAC bounds for how many steps are needed before one can estimate a near optimal policy for any linear reward function.

Strengths: The paper presents a new form of result that is quite interesting. The authors does a good job of explaining the context of other relevant literature.

Weaknesses: While it is not uncommon for this kind of work to not have even illustrative forms of experiments, it would still have been nice to see.

Correctness: I have not properly checked the proofs, but I am not aware of anything incorrect.

Clarity: The paper is well written.

Relation to Prior Work: The authors does a good job of discussing relevant literature.

Reproducibility: Yes

Additional Feedback:

[Author Response · NeurIPS 2020]

We thank the reviewers for their thoughtful reviews; below we address their main concerns. While it only impacts our LSVI analysis and our analysis of FRANCIS remains unchanged, we wish to note that in our own internal re-review we found a small error in the proof of Lemma 23. We apologize for this. Fortunately we can address it by a relatively small change: to define inherent Bellman error using the, perhaps more common, $\infty$-norm, i.e., if eq 2 (def 1) reads

$$\mathcal{I}_{\mathbb{E}} = \max_{Q_{t+1} \in \mathcal{Q}_{t+1}} \min_{Q_t \in \mathcal{Q}_t} \max_{(s,a)} |[Q_t - \mathcal{T}_t^P(Q_{t+1})](s,a)|. \tag{$\star$}$$

This allows us to express the misspecification error (e.g., eqn 37 in appendix) directly in every $(s, a)$ pair, as opposed to in expectation, and the concentration argument in Lemma 23 (which is a concentration argument on deviation of the encountered features wrt their expectation) is no longer needed. Note all the results of the paper continue to hold when using the $\infty$-norm misspecification error in equation $(\star)$ above.

**Explorability (R1, R2, R4)**. We are sorry for the lack of clarity and we are happy to address our explorability assumption and the relation to Chi et al's bounds. Our analysis gives guarantees for the algorithm under two sets of distinct assumptions, which we called *implicit* and *explicit* regularity in the paper (see def. 6 in app.).

**1)** Under *implicit* regularity, we do not put assumptions on the norm of reward parameter $\|\theta^r\|_2$, but only a bound on the expected value of the rewards under any policy: $|\mathbb{E}_{x_t \sim \pi} r_t(x_t, \pi_t(x_t))| \leq \frac{1}{H}$, (see line 762 in appendix). This representation allows us to represent *very high rewards* ($\gg 1$) *in hard-to-reach states*. It basically controls how big the value function can get. This setting is more challenging for an agent to explore *even in the tabular setting* and *even in the case of a single reward function*. If a state is hard to reach, the reward there can be very high, and a policy that tries to go there *can still have high value*. Under this implicit regularity assumption, the explorability parameter would show up for tabular algorithms as well (as minimum visit probability to any state under an appropriate policy): the classical assumption that $|r(s, a)| \leq 1$ would be replaced by $|r(s, a)| \lesssim 1/\nu_{min}$, and the reward / transition noise would become $1/\nu_{min}$ subgaussian; this would ultimately command a corresponding $1/\nu_{min}$ increase in sample complexity even for tabular algorithms *in the fixed reward setting as well as in the reward free setting*. Note that the results from Chi et al. are derived under bounds on the reward parameter which do not satisfy this setting, and therefore our lower bounds are not incompatible with their prior results for the tabular setting.

**2)** Under *explicit* regularity (Definition 6 in the appendix) we do make the classical assumption that bounds the parameter norm $\|\theta^r\|_2 \leq 1/H$ (line 767 in appendix). In this case, our lower bound no longer applies, but the proposed algorithm still requires good "explorability" to proceed (in contrast to, e.g., LSVI-UCB as several reviewers have noticed). We then completely agree with R1, R2 and R4 that the assumption should not be necessary in this case, and removing it is certainly very important, but given the already existing challenges brought by the inherent Bellman error setting we have to leave this as future work. Nonetheless, we would like to point out that explorability does not make the exploration problem trivial. A random policy (i.e., $\epsilon$-greedy) can still take exponential time to learn; other authors [1, 5] have made similar or even stronger assumptions (a bound on the minimum visit probability to any state) to proceed in the context of function approximation.

**Layer-by-layer learning (R4)** As the reviewer suggests, an algorithm that is able to learn all layers together might be more horizon efficient. Interestingly, with our approach we do already achieve the same horizon dependence as the state of the art [3] for *tabular* RL at the time of submission $O(S^2AH^5/\epsilon^2)$ (see also table 1 in our manuscript).

**Motivation (R4)** With this work our aim was to try to weaken the assumptions the current literature makes in the exploration setting with linear function approximation. Although the minimum set of assumptions for RL to work are not well understood even in the linear case [2], our objective is to at least have algorithms that work under assumptions typically made in the batch setting, i.e., when mainstream batch algorithms like least square value (or policy) iteration work [6, 4] using already collected data. We consider this work as a first step in this direction.

**Clarity (R2)** We thank the reviewer for highlighting the clarity issue, and we do plan to use the extra page that is allowed for the camera ready to expand the motivation behind the algorithm, to give a more detailed proof sketch and to clarify the role of explorability, which is especially important as the algorithm style is different than those presented in a number of recent theoretical papers.

# References

[1] S. Du, A. Krishnamurthy, N. Jiang, A. Agarwal, M. Dudik, and J. Langford. Provably efficient RL with rich observations via latent state decoding. In *Proceedings of the 36th International Conference on Machine Learning*, 2019.

[2] S. S. Du, S. M. Kakade, R. Wang, and L. F. Yang. Is a good representation sufficient for sample efficient reinforcement learning? In *International Conference on Learning Representations*, 2019.

[3] C. Jin, A. Krishnamurthy, M. Simchowitz, and T. Yu. Reward-free exploration for reinforcement learning. In *International Conference on Machine Learning (ICML)*, 2020.

[4] M. G. Lagoudakis and R. Parr. Least-squares policy iteration. *Journal of machine learning research*, 4(Dec):1107–1149, 2003.

[5] D. Misra, M. Henaff, A. Krishnamurthy, and J. Langford. Kinematic state abstraction and provably efficient rich-observation reinforcement learning. In *International Conference on Machine Learning (ICML)*, 2020.

[6] R. Munos and C. Szepesvári. Finite-time bounds for fitted value iteration. *Journal of Machine Learning Research*, 9(May):815–857, 2008.


[Meta-Review · NeurIPS 2020]

After considerable discussion, I feel that this paper does meet the acceptance threshold . There needs to be added discussion of the necessity of explorability, since the lower bound is not convincing. Further in the tabular case, explorability is not needed but this algorithm would fail. These should all be carefully discussed.